# MmcA is an electron conduit that facilitates both intracellular and extracellular electron transport in *Methanosarcina acetivorans*

Dinesh Gupta ®[1], Keying Chen[2], Sean J. Elliott ®[2] & Dipti D. Nayak ®[1]✉

Methanogens are a diverse group of Archaea that obligately couple energy conservation to the production of methane. Some methanogens encode alternate pathways for energy conservation, like anaerobic respiration, but the biochemical details of this process are unknown. We show that a multiheme *c*-type cytochrome called MmcA from *Methanosarcina acetivorans* is important for intracellular electron transport during methanogenesis and can also reduce extracellular electron acceptors like soluble $Fe^{3+}$ and anthraquinone-2,6-disulfonate. Consistent with these observations, MmcA displays reversible redox features ranging from −100 to −450 mV versus SHE. Additionally, mutants lacking *mmcA* have significantly slower $Fe^{3+}$ reduction rates. The *mmcA* locus is prevalent in members of the Order *Methanosarcinales* and is a part of a distinct clade of multiheme cytochromes that are closely related to octaheme tetrathionate reductases. Taken together, MmcA might act as an electron conduit that can potentially support a variety of energy conservation strategies that extend beyond methanogenesis.

Methanogens are a polyphyletic group of Archaea that can reduce $CO_2$ or simple organic compounds to methane and couple this metabolic transformation to energy conservation[1,2]. Based on how a chemiosmotic gradient is established, methanogens can be divided into two groups: (a) methanogens without cytochromes and (b) methanogens with cytochromes[3,4]. Methanogens without cytochromes rely on the penultimate step in methanogenesis, mediated by a membrane-bound enzyme called $N^5$-methyl $H_4$MPT: CoM Methyltransferase (Mtr), to generate a $Na^+$ gradient. Mtr catalyzes the exergonic transfer of a methyl group from the $C_1$ carrier tetrahydromethanopterin ($H_4$MPT) to coenzyme M (CoM) and concomitantly pumps out two $Na^+$ ions that can be coupled to ATP generation using a membrane-bound ATP synthase (Supplementary Fig. 1)[3,4]. In contrast, methanogens with cytochromes use an electron transport chain (ETC) to generate an ion gradient for energy conservation. The terminal reductase complex in the ETC of these methanogens is a membrane-bound heterodisulfide reductase that uses the heterodisulfide of coenzyme M (CoM-SH) and coenzyme B (CoB-SH) i.e., CoM-S-S-CoB, generated during the last step of methanogenesis, as the electron acceptor (Supplementary Fig. 2)[3,4].

Taken together, regardless of the underlying mechanism, all methanogens are obligately dependent on methanogenesis to generate an ion motive force for energy conservation and lack any alternate strategies for ATP generation.

Over the years, several studies allude to the possibility that methanogens might be able to decouple ATP generation from methanogenesis and conserve energy by iron respiration using either soluble $Fe^{3+}$ or Fe(III) containing minerals like ferrihydrite as electron acceptors[5–17]. If certain methanogens are indeed capable of switching between methanogenic and non-methanogenic modes of energy conservation, it would dramatically alter our perceived view of their role in the global biogeochemical cycles of macro- and micronutrients. Hence, it is critical to assess iron respiration in methanogens rigorously by identifying specific proteins involved in the process and investigating their function in vivo and in vitro.

There is substantial disagreement in the literature about the taxonomic breadth of methanogens that can perform iron respiration and the resulting physiological and ecological consequences. While initial reports demonstrated iron respiration by methanogens with and

---

[1]Department of Molecular and Cell Biology, University of California, Berkeley, CA, USA. [2]Department of Chemistry, Boston University, Boston, MA, USA. ✉e-mail: dnayak@berkeley.edu

without cytochromes[5,18,19], many recent studies show that this trait is limited to methanogens with cytochromes i.e. members of the Order *Methanosarcinales*[11,12]. That said, some strains within the *Methanosarcinales*, like *Methanolobus vulcani*, cannot reduce soluble $Fe^{3+}$ [5], even in the presence of a humic acid analog - anthraquinone-2,6-disulfonate or AQDS- that typically stimulates iron reduction. Altogether, the distribution of iron respiration across methanogens is strongly but not perfectly correlated with the presence of an ETC. The impact of iron respiration on methanogenesis and cell growth also warrants careful consideration. Typically, the addition of $Fe^{3+}$ and/or AQDS to actively growing cultures of methanogens inhibits methanogenesis and then $Fe^{2+}$ and/or $AQDSH_2$ start to build up, which suggests that these two processes cannot occur simultaneously likely due to some molecular incompatibility[8,11,18,20]. In contrast, some studies[12,17,21] report that $Fe^{3+}$ supplementation enhances methanogenesis, implying that the two processes occur via mutually exclusive pathways. Altogether, despite conflicting evidence, it is clear that certain methanogens are metabolically active under iron-reducing conditions, however the underlying mechanisms warrant further investigation.

In recent studies, the genetically tractable methanogenic archaeon, *Methanosarcina acetivorans*, has been developed as a model system to probe the molecular details of energy conservation coupled to iron respiration. Biochemical assays with purified membranes or everted membrane vesicles show that a membrane-associated multiheme *c*-type cytochrome (MHC), likely MmcA, is involved in the electron relay from reduced ferredoxin to a membrane-bound electron carrier methanophenazine (MP) during methanogenesis and to $Fe^{3+}$ during iron reduction[12,22,23]. Since membrane preps contains many other proteins as well as other *c*-type cytochromes, a direct interaction between MmcA and MP or $Fe^{3+}$ cannot be concluded from these studies[12,22,23]. Other studies show that a mutant lacking *mmcA* is incapable of transferring electrons to AQDS, which further corroborates its involvement in iron reduction[11]. However, the phenotype of this mutant during methanogenic growth on acetate has been shown to vary. The Δ*mmcA* mutant generated by our group[24,25] cannot grow on acetate even after a whole year of incubation whereas another group[11] has reported that this mutant has no growth phenotype during methanogenesis. While multiple lines of investigation have converged on the importance of MmcA in the ETC during methanogenesis as well as iron respiration, the exact role that this protein plays under either condition remains unclear.

In this study, we use a combination of in vitro and in vivo techniques to delineate two distinct biochemical roles for MmcA that depend on the environmental context. First, we devise a strategy to enrich MmcA from *M. acetivorans* and provide strong evidence that it is, as predicted, a membrane-associated heptaheme MHC. Next, we demonstrate that this protein can interact with and transfer electrons to MP as well as AQDS and soluble $Fe^{3+}$ species. The redox properties and evolutionary origins of MmcA further corroborate our proposed functions for this protein. Overall, MmcA is a versatile electron carrier in the ETC of methanogens and can facilitate energy conservation by methanogenesis or iron respiration depending on the availability of oxidized iron species in the environment.

## Results

### MmcA from *M. acetivorans* is a heptaheme *c*-type cytochrome

*MmcA* (locus tag: *MA0658* or *MA_RS03460*) is the first gene of the Rnf (Rhodobacter nitrogen fixation) operon (*MA0658-MA0665* or *MA_RS03460-MA_RS03495*) and contains five canonical (CXXCH) and two non-canonical (CXXXCH and CXXXXCH) heme-binding motifs. The holo-protein is predicted to be a 476 aa-long heptaheme *c*-type cytochrome (cyt *c*) after the signal sequence is processed (Supplementary Fig. 3). MmcA is an essential component of the Rnf complex in *M. acetivorans*[22,24–27] and is primarily found in the membrane fraction (Supplementary Fig. 4). Since MmcA lacks an identifiable transmembrane domain, we anticipate that it is tethered in the membrane through interactions with membrane integral proteins of the Rnf complex[27,28]. To purify MmcA, we generated an *M. acetivorans* strain with an expression vector that encodes a C-terminally tagged (3 × FLAG tag and a twin-Strep tag) *mmcA* placed under the control of tetracycline-inducible promoter[24]. Our previous work shows that the tag does not interfere with maturation and can also rescue the growth defect of the Δ*mmcA* mutant on methylated compounds[24,25]. Since the protein contains a twin-Strep tag, we attempted to purify the protein using a Streptactin resin but were unsuccessful (Supplementary Fig. 5)[29,30]. We then attempted to purify the protein using an anti-Flag resin and were able to obtain ~250 µg of protein per liter of culture (Supplementary Fig. 6). We performed a peroxidase-based heme stain and an immunoblot with anti-Flag antibody to confirm that the protein fraction is primarily MmcA and contains covalently bound heme groups i.e. is a holo-form of MmcA (Fig. 1A). The MmcA enriched fraction has a distinct red color which is a hallmark of cyt *c* (Fig. 1A) and also has spectral features typical of cyt *c* with an absorption maximum at 408 nm (γ) and 530 nm in the oxidized state (in black) and 419.5 nm (γ), 523 nm (β), and 552 nm (α) in the reduced state (in red) (Fig. 1B). We also confirmed the presence of covalently attached heme groups in MmcA by performing a pyridine hemochrome assay and observed a

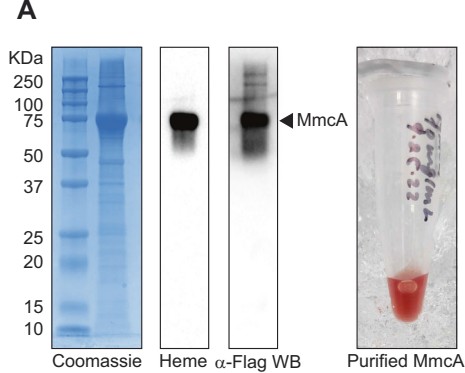

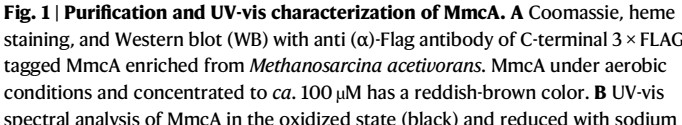

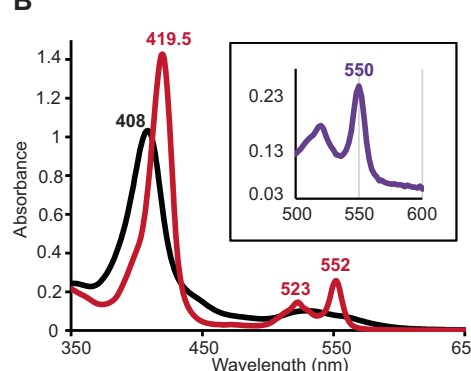

**Fig. 1 | Purification and UV-vis characterization of MmcA. A** Coomassie, heme staining, and Western blot (WB) with anti (α)-Flag antibody of C-terminal 3 × FLAG tagged MmcA enriched from *Methanosarcina acetivorans*. MmcA under aerobic conditions and concentrated to *ca.* 100 µM has a reddish-brown color. **B** UV-vis spectral analysis of MmcA in the oxidized state (black) and reduced with sodium dithionite (red). The inset shows the pyridine hemochrome assay of reduced MmcA with a characteristic alpha peak for *c*-type cytochromes at 550 nm. (Complete UV-vis spectra of the hemochrome assay is shown in Supplementary Fig. 7). Data shown in (**A**) and (**B**) are representative of three experiments (*n* = 3). Source data are provided as a source data file.

characteristic 550-nm α peak (Fig. 1B, inset and Supplementary Fig. 7). Taken together, these data show that MmcA from *M. acetivorans* is a heme-attached cyt *c* protein.

Although MmcA is predicted to be a heptaheme cyt *c*, the heme occupancy of this protein has not been experimentally validated. To this end, we performed LC-MS/MS analysis of chymotrypsin-digested fragments of MmcA and observed that the seven peptides with the putative heme-binding motifs had an increase in mass of 615.17 Da corresponding to heme attachment (Table 1)[31]. This observation confirms that MmcA is a heptaheme cyt *c*, and also demonstrates that the cyt *c* maturation (CCM) machinery of *M. acetivorans* can covalently attach heme to non-canonical heme-binding motifs in MmcA (Supplementary Fig. 3)[24]. We did not detect the predicted signal peptide in either trypsin or chymotrypsin-digested fragments of MmcA, which suggests that it gets cleaved after membrane translocation (Supplementary Fig. 8).

### MmcA is a methanophenazine reductase

Using purified membranes from *M. acetivorans*, it has been shown that MmcA is involved in transferring electrons from ferredoxin to the membrane-bound electron carrier MP via the Rnf complex[22]. However, whether MmcA interacts with MP directly or indirectly via another protein remains unknown. We conducted spectroscopic analyses with MmcA to measure its ability to donate electrons to MP (Fig. 2A). We added an excess of 2-hydroxyphenazine (2HP) (200 μM), a well-established soluble analog of MP[22,32–35], to MmcA reduced with sodium dithionite under anoxic conditions (Fig. 2B). The addition of 2HP

instantaneously oxidized MmcA as observed by a shift in its Soret peak from 419.5 nm to 408 nm and the disappearance of the α and β peaks of reduced MmcA (Fig. 2B; Supplementary Fig. 9). Since the reaction was instantaneous, we were unable to calculate kinetic parameters. Regardless, these data provide strong evidence that MmcA can interact with and donate electrons to MP and establishes a clear role for this protein in the ETC during methanogenesis.

### MmcA can donate electrons to extracellular electron acceptors like ferric iron and humic acid analogs

Previous studies have shown that a mutant of *M. acetivorans* lacking *mmcA* fails to reduce AQDS[11], but these genetic data do not imply that MmcA is directly involved in the reduction of AQDS and if it can also reduce other redox-active molecules (Fig. 3A). We tested if MmcA can reduce AQDS or $Fe^{3+}$ [either ferric chloride ($FeCl_3$) or ferricyanide ($K_3[Fe(CN)_6]$)] in vitro by monitoring the redox state of MmcA after adding an excess of each of these electron acceptors (Fig. 3). All three electron acceptors oxidized MmcA rapidly as confirmed by the change in the spectral profile of the protein (Fig. 3; Supplementary Figs. 10 and 11). In contrast, a control experiment with $Zn^{2+}$ as the sole electron acceptor did not change the redox state of dithionite-reduced MmcA (Supplementary Figs. 12 and 13). While we were unable to measure the reaction kinetics using our experimental setup, these data confirm the interaction between MmcA and AQDS or $Fe^{3+}$ and validate its important role during anaerobic respiration in *M. acetivorans*.

### MmcA enhances fitness under iron-reduction conditions in *M. acetivorans*

To test the role of MmcA in vivo, we measured the rate of $Fe^{3+}$ reduction with methanol as the electron donor in live cell suspensions of *M. acetivorans* (Fig. 4A–C). Consistent with our expectations, the absence of *mmcA* slows down $Fe^{3+}$ reduction rates by ~30% in vivo (Fig. 4B, C). To test for polar effects of deleting *mmcA* on the chromosome, we measured the rate of iron reduction in the Δ*mmcA* mutant complemented with *mmcA* on a plasmid. Complementing *mmcA* in *trans* restored iron reduction rates to wild-type levels (Fig. 4B, C). These data suggest that MmcA improves fitness of *M. acetivorans* in $Fe^{3+}$-containing environments.

### MmcA is reversibly redox active between −100 and −450 mV versus SHE

We explored the redox behavior of MmcA using protein film voltammetry (PFV) on the meso-porous indium tin oxide electrode

**Table 1 | Chymotrypsin-digested MmcA peptides with heme-binding motifs**

| MmcA peptides with heme-binding motif | Observed mass | Expected mass | Difference |
|---|---|---|---|
| SDSVCGGCHFGVY | 1945.689 | 1330.5191 | 615.1699 |
| GIDIDCMMCHEKY | 2172.793 | 1557.6205 | 615.1725 |
| HDAVNGAPISCAQRCHRIDVETSAVMW | 3581.5553 | 2966.3818 | 615.1735 |
| ADEEDFEESDAHAANGVECTECHHTEAF | 3708.3508 | 3093.1745 | 615.1763 |
| DDTMRSCDDAECHAGISHGPF | 2879.0504 | 2263.8801 | 615.1703 |
| LACEACHTPELPGGDLPGGNVL | 2778.1924 | 2163.0209 | 615.1715 |
| TCKDCHGNEAVIDW | 2205.8406 | 1590.6675 | 615.1731 |

Heme-binding motifs (CXXCH, CXXXCH and CXXXXCH) are shown in red. For all these peptides a mass shift of 615.17 Dalton corresponding to heme c was observed.

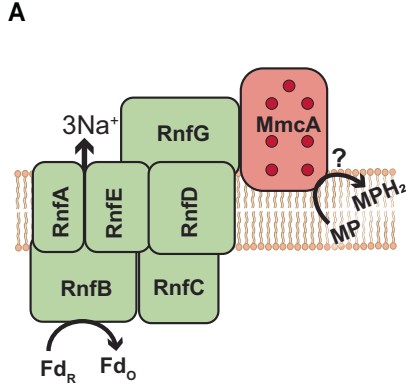

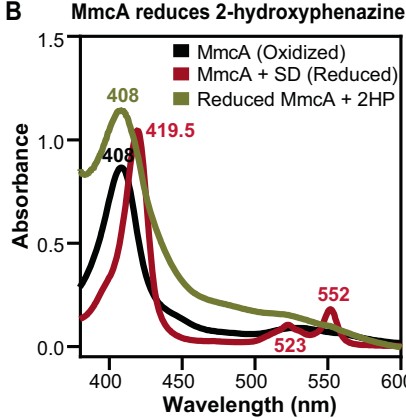

**MmcA reduces 2-hydroxyphenazine**

**Fig. 2 | MmcA can function as a methanophenazine (MP) reductase.**
**A** Schematic of the MmcA-Rnf complex (adapted from Gupta et al. [24]) where the multiheme cytochrome MmcA mediates the final step in the transfer of electrons from the complex to the membrane-bound electron carrier methanophenazine (MP). **B** Spectral analysis of MmcA-mediated reduction of the soluble MP analog, 2-hydroxyphenazine (2HP) under anaerobic conditions. Addition of 2HP to MmcA reduced with sodium dithionite (SD; red) leads to the oxidation of MmcA (olive green) as observed by the appearance of a characteristic Soret peak at 408 nm indicative of the oxidized protein (black). Data shown are representative of two experiments ($n = 2$). Source data are provided as a source data file.

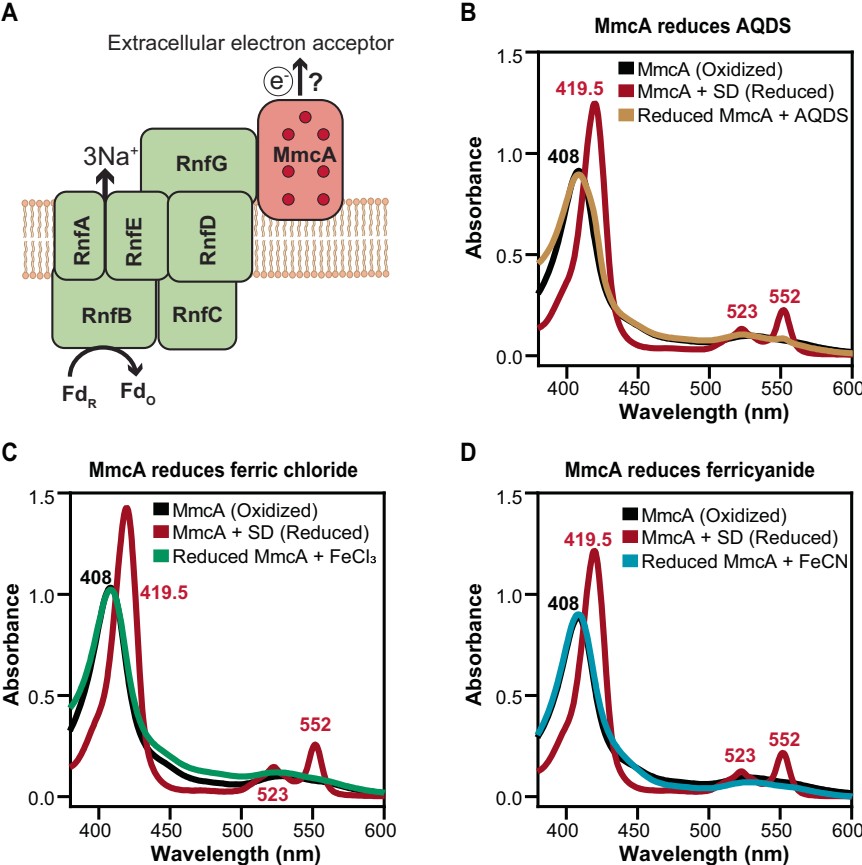

**Fig. 3 | MmcA can donate electrons to extracellular electron acceptors.**
**A** Schematic of the MmcA-Rnf complex (adapted from Gupta et al. [24]) where the multiheme cytochrome MmcA mediates the final step in the transfer of electrons from the complex to extracellular electron acceptors. **B–D** Spectral analysis of MmcA-mediated reduction of anthraquinone-2,6-disulfonate (AQDS) (**B**), ferric chloride (**C**), and ferricyanide (**D**) under anaerobic conditions. Addition of AQDS (**B**), ferric chloride (**C**), and ferricyanide (**D**) to MmcA reduced with sodium dithionite (SD; red) leads to the oxidation of MmcA (yellow in **B**; green in **C**; blue in **D**) as observed by the appearance of a characteristic Soret peak at 408 nm indicative of the oxidized protein (black). Data shown are representative of two experiments ($n = 2$). Source data are provided as a source data file.

(ITO). MmcA formed a stable film on the ITO surface and was capable of exchanging electrons directly with the electrode, giving rise to reversible redox signals spanning from −100 to −450 mV (Fig. 5A; Supplementary Figs. 14 and 15). These signals, once deconvoluted and fit to Nernstian one-electron peaks, could be separated into seven reversible signals corresponding to the reduction and oxidation of the seven heme cofactors within MmcA, with uncertainties of approximately 10 mV. We determined the midpoint potentials of each heme by taking the average of the reduction and oxidation peak potentials for each redox pair (Fig. 5B). The low potential redox range of MmcA is similar to functionally analogous MHCs like MtrA and the MtrCAB complex (0 to −400 mV and 0 to −450 mV, respectively) in *Shewanella oneidensis* MR-1[36,37] and OmcZ or OmcS (−60 to −420 mV and −40 to −360 mV, respectively) in *Geobacter sulfurreducens*[38,39]. The redox-active range of MmcA suggests that it is capable of transferring electrons to $Fe^{3+}$ (+300 to +400 mV), AQDS (−185 mV), and MP (−165 mV) (Fig. 5B).

**MmcA is related to the tetrathionate reductase (OTR) family of multiheme *c*-type cytochromes**
The *mmcA* locus is only present in methanogens and anaerobic methane-oxidizing archaea (ANME) within the Order *Methanosarcinales* and has *ca.* 25% amino acid sequence similarity to putative octaheme tetrathionate reductases (OTR) in other Archaea within the Orders *Methanosarcinales*, *Desulfurococcales*, and *Archaeoglobales*. A rooted tree of MmcA and OTR sequences shows

that the MmcA clade is distinct but closely related to OTRs from Archaea as well as Bacteria (Fig. 6A). These observations support an independent origin for MmcA from an ancestor of MmcA and OTR rather than MmcA being derived from OTRs within Archaea. This hypothesis is further supported by synteny analyses of *mmcA* and *otr* in methanogens that encode both loci. While *mmcA* is always found in an operon with other genes of the Rnf complex, *otr* is present at a distant locus, often near a thermosome subunit and a biotin transporter (Fig. 6B).

Unlike MmcA, OTR is a respiratory enzyme of cryptic function, best characterized as a reductase of tetrathionate, nitrite, and hydroxylamine in bacteria like *Shewanella oneidensis*[40,41]. All OTRs, including those derived from members of the *Methanosarcinales*, contain eight heme-binding motifs and structural studies show that the second heme group, involved in catalysis, is ligated by a highly conserved Lys instead of His[40]. Neither this heme-binding motif nor the Lys are conserved in the MmcA sequence or predicted structure (Fig. 6C, D and Supplementary Fig. 16). All seven heme-binding motifs in MmcA are within electron transfer range for the heme groups with a total head-to-head distance of approximately ~55 Å between the first to seventh heme groups (Fig. 6E). Finally, the range of the redox potentials of the heme groups in MmcA (Fig. 5) is similar to that of OTR derived from *S. oneidensis* (Fig. 5B, Supplementary Fig. 17). Altogether, these data suggest that the MmcA clade of MHCs is distinct in form and function but closely related to the well-characterized OTR family of MHCs.

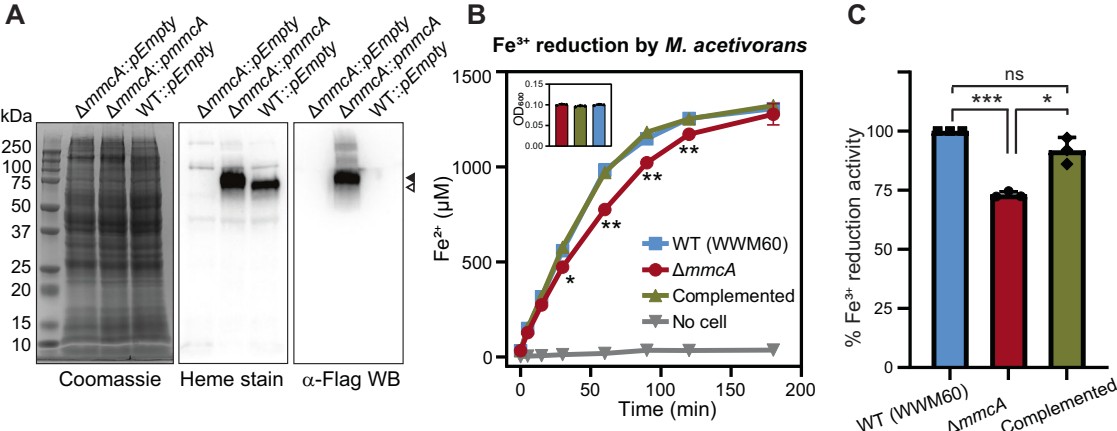

**Fig. 4 | MmcA facilitates iron reduction in *M. acetivorans*. A** Coomassie, heme stain, and Western blot (WB) with anti (α)-FLAG antibody of cell lysates obtained from a Δ*mmcA* mutant complemented with a plasmid encoding a C-terminal 3 × FLAG tagged *mmcA* placed under the control of a tetracycline-inducible promoter (Δ*mmcA::pmmcA*). The controls are either the Δ*mmcA* mutant or the parent strain (WWM60; wild type or WT) with an empty vector (pDPG010; labeled here as *pEmpty*) as described before[24]. Wild-type MmcA and C-terminal 3 × FLAG tagged MmcA are depicted by open and filled arrows respectively. **B** The rate of $Fe^{3+}$ reduction in 1 mL anaerobic cell suspensions with similar optical density (see inset, the bar diagram represents the mean with standard deviation of three technical replicates, $n = 3$) was determined by an increase in $Fe^{2+}$ using the ferrozine assay. 100 µg/mL of tetracycline was used to induce expression. Error bars are means ± standard deviation of three technical replicates ($n = 3$). Asterisks represent *p*-values

of Δ*mmcA* compared to the complemented strain and are only shown when it is also statistically significant for Δ*mmcA* compared to WT (*p*-values for Δ*mmcA* compared to WT & Δ*mmcA* compared to complemented strain at time points 30, 60, 90, and 120 min are 0.0489 & 0.0176, 0.0294 & 0.0046, 0.0019 & 0.0099, and 0.0225 & 0.0088, respectively). **C** Rate of $Fe^{3+}$ reduction by the Δ*mmcA* and Δ*mmcA::pmmcA* complementation strain were calculated as the percentage activity of the parent strain (WWM60) denoted as wild type (WT). The data shown are average ± standard deviation from three independent experiments ($n = 3$) done with three technical replicates ($n = 3$) each. **p*-value ≤ 0.05, ***p*-value ≤ 0.01, ****p*-value ≤ 0.001 and ns is *p*-value ≥ 0.05 respectively using a two-sided Student's *t*-test. The *p*-values are 0.00079, 0.019, and 0.124 for iron reduction rates compared to Δ*mmcA* vs WT, Δ*mmcA* vs complemented strain, and complemented strain vs WT, respectively. Source data are provided as a source data file.

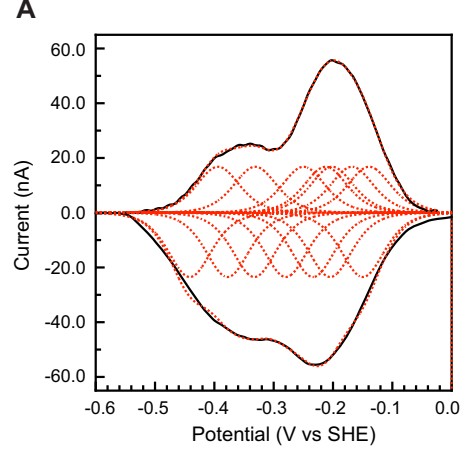
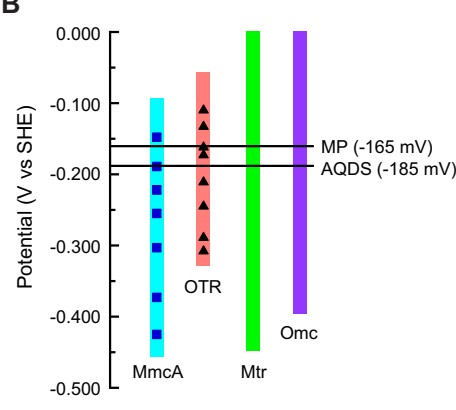

**Fig. 5 | MmcA is redox active at a potential between −100 and −450 mV versus SHE. A** Background-subtracted non-turnover MmcA voltammogram (black solid) with fitting of seven reversible redox couples (red dotted). Cyclic voltammetry (CV) was performed at pH 7.4, 10 °C, and a scan rate of 20 mV/s. **B** Redox range of MmcA from *Methanosarcina acetivorans* and octaheme tetrathionate reductase (OTR) from *Shewanella oneidensis* (See Supplementary Fig. 17). The average midpoint potentials ($E_m$) of the seven heme centers in MmcA (blue

squares) and eight heme centers in OTR (black triangles) are calculated from three experimental replicates ($n = 3$) under identical conditions. The redox range of other multiheme *c*-type cytochromes like the metal reductase (Mtr) from *Shewanella oneidensis*[36,37] and the outer membrane cytochromes (Omc) from *Geobacter sulfurreducens*[38,39] as well as the midpoint potential of methanophenazine (MP) and anthraquinone-2,6-disulfonate (AQDS) are shown for reference.

## Discussion

In principle, if methanogens with cytochromes could reconfigure their ETC to also conserve energy using extracellular electron acceptors like iron, they would survive (if not proliferate) in far more diverse ecological niches (Supplementary Table 1). In practice, this phenomenon has been demonstrated repeatedly using environmental samples as well as axenic cultures, but biochemical details of the molecular mechanism(s) involved are yet to emerge. In the absence of any extracellular electron acceptors, *M. acetivorans* performs methanogenesis to generate a heterodisulfide, CoM-S-S-CoB, which serves as the terminal

electron acceptor for the ETC[3]. Under these circumstances, MmcA transfers electrons from the Rnf complex to the membrane-bound electron carrier MP (Supplementary Fig. 2; Fig. 2), which can ultimately be used to reduce the terminal electron acceptor CoM-S-S-CoB. MmcA is universally conserved in methanogens with an Rnf complex and absent in Bacteria that use the Rnf complex to transfer electron between the ferredoxin and NAD pools, further corroborating its unique role as an electron conduit between the Rnf complex and MP[28]. When *M. acetivorans* encounters soluble $Fe^{3+}$ or AQDS, our data suggest that MmcA can directly interact with and reduce these electron

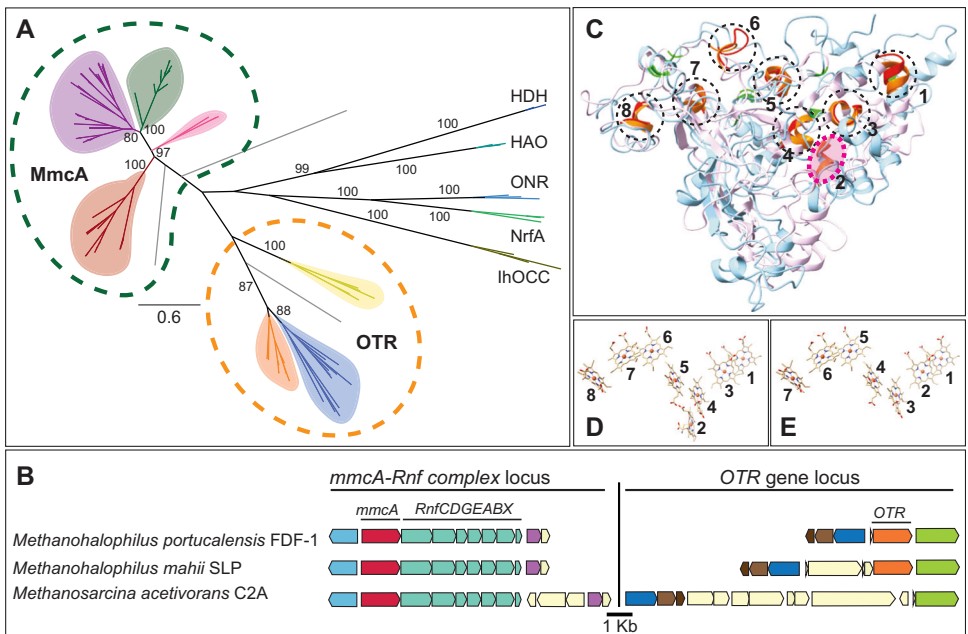

**Fig. 6 | MmcA comprises a distinct clade of multiheme cytochromes related to the octaheme tetrathionate reductases (OTR). A** Maximum-likelihood phylogenetic tree of MmcA and representatives of the OTR family. MmcA clades are colored by members of the Genus *Methanosarcina* (green), other members of the Family *Methanosarcinaceae* (purple), unspecified MAGs from the Order *Methanosarcinales* (pink), and anaerobic methane-oxidizing archaea (ANME) (red). OTR clades are colored by members of the *Methanosarcinaceae* (orange), members of the *Desulfurococcales* and *Archaeoglobales* (yellow), and representatives from Bacteria (blue). A pentaheme nitrite reductase (NrfA), octaheme nitrite reductase (ONR and IhOCC), octaheme hydrazine dehydrogenase (HDH) and octaheme hydroxylamine oxidoreductase (HAO) derived from bacteria were used as the outgroup to root the tree. Bootstrap values of 80 and above are shown.

**B** Chromosomal organization of genes surrounding the *mmcA* locus (red) and the *otr* locus (orange) in a few representative strains within the Family *Methanosarcinaceae*. Genes of the same color represent members of the same orthologous group. **C** Structural alignment of AlphaFold predicted model of signal-less MmcA (MA0658; 25–500 aa, cyan) to crystal structure of OTR from *Shewenella oneidensis* (PDB: 1SP3, pink). Heme-binding motifs of OTR (orange) and MmcA (red) are shown. The histidine (His) ligand of the seven bis-His coordinated heme groups (1, 3–8) of OTR (light green) and seven heme groups (1–7) of MmcA (dark green) are shown. The second heme-binding motif of OTR that is absent in MmcA is highlighted (pink circle). Arrangement of heme groups in the OTR crystal structure (**D**) and in the MmcA model (**E**).

acceptors in vivo and in vitro (Figs. 3 and 4). Here, it is worth clarifying that even though we and others[12,23] have shown that methanogens like *M. acetivorans* are metabolically active and can conserve energy by iron respiration (Fig. 4), robust growth that spans multiple generations is yet to be demonstrated i.e. it is still not known whether methanogens can couple iron reduction to growth in addition to energy conservation. Regardless, redox transformation of iron species by methanogens has substantial biogeochemical ramifications in and of itself to merit further investigation.

Methanogenic archaea like *M. acetivorans* typically lack a cell wall and their cell envelope comprises of an inner membrane and a crystalline proteinaceous surface layer (S-layer)[42]. While MmcA is membrane-associated (Supplementary Fig. 4), it lacks any S-layer domains, suggesting that it is present in the pseudo-periplasm rather than the outer-surface of cells[28]. The hexagonal crystal packing of the S-layer forms three different pores of which one, the primary pore, is large enough to allow the passage of chelated metal ions and AQDS into the pseudo-periplasm[43]. As a result, it is entirely feasible that MmcA can interact with many soluble extracellular electron acceptors despite its pseudo-periplasmic localization. While some recent studies suggest that MmcA contributes substantially to electron uptake from metallic iron[44], more work needs to be done in the future to determine the involvement of MmcA in direct electron transfer to and from minerals and electrodes.

Even though MmcA plays an important role in $Fe^{3+}$ reduction, our in vivo data clearly suggest that *M. acetivorans* encodes additional routes for extracellular electron transport (EET) (Fig. 4). Redundancy in EET pathways seems to be a common phenomenon in other well-studied microorganisms too. Deleting any one of five MHCs (OmcS,

OmcZ, OmcB, OmcT, and OmcE) has little to no effect on AQDS reduction in *Geobacter sulfurrreduces*, and only a quintuple mutant lacking all five genes is unable to reduce AQDS and humic acids[45]. *M. acetivorans* encodes at least four other cyt *c* and the expression of one of these genes (MA3739) increases by 80% in the Δ*mmcA* mutant (Supplementary Table 2), suggesting that other cyt *c* might be able to functionally complement MmcA in its absence. Alternately, it is entirely plausible that other pathway(s) for iron respiration are MHC-independent as members of the *Methanosarcinales* lacking cyt *c* have also been shown to reduce iron or participate in direct interspecies electron transfer (DIET)[5,8–10,46]. Even though it is not technically feasible at present, high-throughput $Fe^{3+}$ reduction assays with an unbiased transposon mutant library of the Δ*mmcA* strain might help identify additional pathways for EET in the future.

In conclusion, we show that MmcA is an important constituent of the respiratory chain of methanogens like *M. acetivorans*. Notably, based on the availability of electron acceptors like $Fe^{3+}$ or AQDS, MmcA might also facilitate energy conservation by anaerobic respiration in addition to methanogenesis, thus broadening the ecological niche of these pivotal organisms.

## Methods
### Growth medium
*M. acetivorans* strains (Supplementary Table 3) were grown in single-cell morphology[47] at 37 °C without shaking in bicarbonate-buffered high-salt (HS) liquid medium containing either methanol or trimethylamine (TMA) as the carbon and energy substrate with $N_2/CO_2$ (80/20) in the headspace. Puromycin (RPI, Mount Prospect, IL) was added to a final concentration of 2 μg/mL from a sterile, anaerobic stock solution

to select for *M. acetivorans* strains with the *mmcA*-expression plasmid encoding a puromycin-resistance gene (*pac*). Anaerobic, sterile stocks of tetracycline hydrochloride in deionized water were prepared fresh before use and added to a final concentration of 100 μg/mL to induce the expression of MmcA from the tetracycline-inducible promoter as described previously in ref. [48]. Cell cultures with a volume of up to 10 mL were grown in Balch tubes and larger volume cultures were grown in anaerobic bottles.

## MmcA purification

MmcA was enriched by affinity purification from 4 L of late-exponential phase culture of DDN039 grown in HS media with 100 mM TMA at 37 °C. Cells were harvested by centrifugation (6000 × *g*) for 20 min at 4 °C, the supernatant was discarded, and the cell pellets were stored at −80 °C. All steps of protein purification were performed under aerobic conditions. 2 U/mL DNase-I (to reduce the viscosity of the suspension) and 1 mM Phenylmethylsulfonyl fluoride (PMSF) (to inhibit protease activity) was added to 20 mL of hypotonic lysis buffer (50 mM Tris-HCl, pH = 7.4) used to resuspend the cell pellet. The cell suspension was kept on ice for 45 min with intermittent mixing using a pipette to lyse the cells. Upon complete lysis, sodium chloride was added from a 5 M stock solution to a final concentration of 150 mM to the cell lysate. The lysate was clarified by centrifugation at 10,000 × *g* for 20 min at 4 °C and the supernatant was separated into the soluble and membrane fractions by high-speed ultracentrifugation at 100,000 × *g* for 1 h at 4 °C. The membrane pellets were solubilized in 4 mL TBS buffer (50 mM Tris-HCl, 150 mM NaCl, pH = 7.4) with 2% Triton X-100 (Sigma−Aldrich, St Louis, MO, USA). The solubilized membrane fraction was loaded on a column containing 1 mL anti-DYKDDDDK (Flag) G1 affinity resin (50% suspension; GenScript, Piscataway, NJ, USA) pre-equilibrated with 3 bed volumes of TBS buffer. Five washes with 2 mL of TBS buffer were performed before the protein was eluted using competitive elution buffer (300 μg/mL Flag peptide in TBS buffer). To elute the protein, three times the bed volume (i.e., 1.5 mL) of elution buffer was added to the column and one volume (500 μL) of elute was collected right away. The column was capped and incubated at room temperature for 30 min before collecting the rest of the eluate. The elutes were quantified using Bradford reagent (Sigma−Aldrich, St Louis, MO, USA) with BSA (bovine serum albumin) as the standard following the manufacturer's instructions and saved at −80 °C.

## Heme staining and Western blot

Peroxidase-based assays for heme staining were performed as described previously[24,49]. Briefly, MmcA containing samples or total cell lysate was mixed with loading dye (without β-mercaptoethanol), incubated at 65 °C for 4 min, and resolved by running 12% Mini-Protean TGX SDS-PAGE gels (Bio-Rad, Hercules, CA, USA). Gels were transblotted to 0.2 μm PVDF membrane (Bio-Rad, Hercules, CA, USA) using Trans-Blot Turbo transfer system (Bio-Rad, Hercules, CA, USA) and developed with SuperSignal West Femto kit (Thermo Scientific, Waltham, MA, USA) to detect the heme signal. For Western blot, the heme stain blots were treated with 50 mL stripping buffer (60 mM Tris pH = 7 containing 2% SDS and 7 μL/mL β-mercaptoethanol) shaking at 50 rpm for 1 h at 50 °C. After confirming for the absence of any peroxidase-based signal from heme, the presence of MmcA with FLAG tag was probed with immunoblotting using monoclonal anti-Flag M2-Peroxidase (HRP) antibody (Cat # A8592, Lot # SLCF0816, Sigma−Aldrich, Saint Louis, MO, USA) (1/60,000X dilution) and developed with Immobilon Western Chemiluminescent HRP Substrate (Millipore, Burlington, MA, USA) for signal detection. The ChemiDoc MP Imaging System (Bio-Rad, Hercules, CA, USA) was used for imaging. Protein concentrations were estimated using the Bradford reagent (Sigma−Aldrich, Saint Louis, MO, USA) with BSA (bovine serum albumin) as the standard per the manufacturer's instructions.

## Proteolytic digestion of MmcA and LC-MS/MS analysis

MmcA was digested with trypsin or chymotrypsin in solution for LC-MS/MS analysis as previously described[50]. Briefly, 20 μL of MmcA at a concentration of 2 mg/mL (total 40 μg) was mixed with 9.6 mg of urea (8 M) in a sterile microfuge tube and incubated for 1 h at room temperature for protein denaturation. The MmcA-Urea mix was diluted 10-fold by adding 180 μL freshly prepared 50 mM ammonium bicarbonate solution. Two aliquots of 100 μL were transferred into new tubes and digested either with a sequencing-grade trypsin (Promega, Madison, WI, USA) or chymotrypsin (Promega, Madison, WI, USA) per manufacturer's instructions. For trypsin digestion, 2.5 μL of 0.4 μg/μL trypsin (1 μg) was added to 100 μL protein solution (1:20, enzyme to protein ratio) and incubated at 37 °C overnight (*ca.* 16 h). Similarly, 2 μL of 0.5 μg/μL chymotrypsin (1 μg) was added to another 100 μL protein solution (1:20, enzyme to protein ratio) and incubated at 25 °C overnight (18 h). 75 μL of the overnight-digest were transferred to a clean microfuge tube and submitted for MS analysis (QB3/Chemistry Mass Spectrometry Facility, UC Berkeley). Two independently purified MmcA samples (*n* = 2) were digested either with trypsin or chymotrypsin and analyzed by Mass Spectrometry. Protein digests were also confirmed by running the remaining 25 μL sample on SDS-PAGE gels followed by Coomassie staining.

## UV-visible (vis) absorption spectroscopy with MmcA

All UV-vis spectroscopy was performed at room temperature with a Shimadzu 1900i (Shimadzu, Torrance, CA, USA) kept inside an anaerobic chamber (97% $N_2$, and 3% $H_2$; Coy Laboratory, Grass Lake, USA). Unless specified, all assays were conducted with 2.5–3.5 μM MmcA in 50 μL of assay buffer (50 mM Tris-HCl, 150 mM NaCl, 2% glycerol, pH = 7.4). Stock solutions of 1 mM and 10 mM sodium dithionite were prepared in deionized water. 2-hydroxyphenazine was custom synthesized (AstaTech Inc., Bristol, PA), and a 20 mM stock solution was prepared in 100% ethanol. A 10 mM stock of Anthraquinone-2,6-disulfonate (AQDS) was prepared in deionized water as described before[51]. The solution was heated at 60 °C until AQDS was completely dissolved (-10 min), cooled down to room temperature, and the pH was adjusted to 7.0. Stocks of 10 mM ferric chloride ($FeCl_3$) and potassium ferricyanide ($K_3[Fe(CN)_6]$) were prepared with deionized water. All assay components (buffer, protein, chemicals) were kept in the anaerobic chamber in small volumes (30−50 μL) at least 2 h prior to the assay and were confirmed to be anaerobic by using the redox dye resazurin (0.0001% w/v) and testing for a color change from colorless to pink after 10 min.

## Pyridine hemochrome assay

This assay was performed as described before[52,53]. Briefly, a 0.2 M NaOH with 40% pyridine solution was made fresh using a 1 M NaOH stock and 100% pyridine solution (Sigma−Aldrich, St. Louis, MO). 5 μL (i.e., 1/200) of 0.1 M potassium ferricyanide stock solution was added to 495 μL of the aforementioned NaOH + pyridine mix to generate the pyridine hemochrome assay solution. 50 μL of the assay solution was mixed with 50 μL of TBS buffer (50 mM Tris-HCl, 150 mM NaCl, pH = 7.4) and used as a blank. Next, 50 μL of the assay solution was mixed with 50 μL of MmcA in TBS buffer, and UV-vis scans were immediately performed using a Shimadzu 1900i (Shimadzu, Torrance, CA, USA) to record the oxidized spectra. A 10 mM stock solution of sodium dithionate was added to the protein assay mixture and UV-vis scans were performed using a Shimadzu 1900i (Shimadzu, Torrance, CA, USA) to record the fully reduced pyridine hemochrome spectra.

## Cell suspension assays

Cell suspension assays were performed in an anaerobic chamber (97% $N_2$, and 3% $H_2$; Coy Laboratory, Grass Lake, USA) at room temperature as previously described for bacterial cells[54] with some modifications. Assays were performed with an *M. acetivorans* mutant lacking the

chromosomal copy of the *mmcA* locus and expressing *mmcA* from an inducible promoter on a plasmid and a control strain containing the plasmid pDPG010 described previously[24]. All strains were grown in HS medium with 125 mM methanol, 2 μg/mL puromycin, and *mmcA* expression was induced by adding tetracycline to a final concentration of 100 μg/mL. Cells were harvested in mid-exponential phase ($OD_{600}$ = 0.4–0.6) by centrifugation in the anaerobic chamber. The cell pellet was resuspended in anaerobic high-salt PIPES buffer (50 mM PIPES, 400 mM NaCl, 13 mM KCl, 54 mM MgCl₂, and 2 mM CaCl₂, pH 6.8) containing 5 mM methanol and washed three times. At the end of the third wash, cells were resuspended in anaerobic high-salt PIPES buffer containing 5 mM methanol and supplemented with a freshly prepared 1:1 $Fe^{3+}$- nitrilotriacetic acid (NTA) mix using an anaerobic 0.4 M ferric chloride and an anaerobic 0.8 M NTA stock solutions to final concentration of 1 mM ferric chloride and 2 mM of NTA. $Fe^{3+}$ reduction was monitored by sampling the suspension at different time points and measuring the $Fe^{2+}$ concentration by using the ferrozine assay[55].

### Electrochemistry
Protein film voltammetry (PFV) experiments were carried out using a three-electrode cell configuration with the cell thermostated at 10 °C and housed inside a nitrogen-filled MBraun Labmaster glovebox (residual O₂ < 1 ppm). The reference electrode was a saturated calomel electrode (SCE), and the counter electrode a platinum wire. The working electrode was a meso-porous indium tin oxide (ITO) electrode, prepared according to reported procedures[56]. Briefly, a pyrolytic graphite edge (PGE) electrode (3 mm diameter) was polished and sonicated in water. ITO nanoparticles (Sigma, <50 nm) were then deposited on the PGE surface by electrophoretic deposition. The PGE electrode was submerged in an acetone solution (20 mL) of I₂ (0.01 g) and ITO (0.02 g), and a potential of 10 V was applied for 6 min using a graphite rod as the auxiliary electrode that was held approximately 1 cm away. The ITO electrode was then thoroughly rinsed with water and dried prior to use. To deposit the protein, a 3 μL aliquot of the protein solution (100 μM MmcA or 55 μM OTR), the latter prepared as previously described[57] was placed on the electrode for 3 min. Excess protein solution was then removed by rinsing with cold buffer, and the electrode was immediately placed into the electrolyte buffer, which contained 10 mM MES, 10 mM MOPS, 10 mM TAPS, 10 mM CHES, 10 mM HEPES, 10 mM CAPS, and 200 mM NaCl (pH 7.4). Cyclic voltammograms (CVs) were collected using the GPES software package (Ecochemie) that was connected to a PGSTAT30 AutoLab potentiostat (Ecochemie). All PFV data were analyzed using the qSOAS package[58], through which background electrode capacitance was subtracted, and data were filtered to remove electrical noise. Deconvolution of the redox feature was achieved within qSOAS using procedures reported previously, where the redox stoichiometry of all 7 heme cofactors was set to 1.0 (*n* = 1).

### Bioinformatics analyses
MmcA homologs were extracted from the NCBI non-redundant protein database using the MmcA (MA0658) protein sequence from *M. acetivorans* as the query and Archaea as the search database. Alignments and tree building were conducted in Geneious Prime 2023.0.3 (https://www.geneious.com). Any partial sequences (<375 aa) were discarded. The sequences were aligned using MUSCLE with default parameters. Maximum-likelihood tree of MmcA was generated using RAxML (Protein Model - GAMMA BLOSUM62; Algorithm - Rapid Bootstrapping and search for best-scoring ML tree; Number of starting trees or bootstrap replicates - 100; Parsimony random seed - 1). Gene ortholog neighborhood analysis was performed on Integrated Microbial Genomes and Microbiomes platform[59]. For structural alignment and model building, an AlphaFold2 prediction of MmcA[60] and its closest available crystal structure i.e., 1SP3 for OTR from *Shewanella*

*oneidensis* (SO4144)[40], were docked using matchmaker tools and 1SP3 as a reference structure in ChimeraX[61].

### Reporting summary
Further information on research design is available in the Nature Portfolio Reporting Summary linked to this article.

## Data availability
The data that support this study are available from the corresponding authors upon request. Source data for Fig. 1A, 1B, 1B-inset; Fig. 2B; Fig. 3B–D; Fig. 4A and B and for Supplementary Figs. 4, 5A, 5B, 6, 7, 9–13 are provided as a Source Data File. 1SP3 was used for structural comparison in Fig. 6C. All the strains used in the study are listed in the supplementary table and will be made available upon request to the corresponding author. Source data are provided with this paper.

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

## Acknowledgements

We would like to acknowledge Dr. Anthony Iavarone for LC/MS analyses of peptide fragments of MmcA, Prof. Donald Rio for access to an ultracentrifuge for protein purification, Dr. Catarina Paquete for the donation of the *Shewanella oneidensis* strain used to produce OTR, Daniel Tekverk for producing OTR, and to members of the Nayak lab for their feedback and input on the manuscript. The authors acknowledge funding from the 'New Tools for Advancing Model Systems in Aquatic Symbiosis' program from the Gordon and Betty Moore Foundation (GBMF#9324 to D.D.N. and D.G.). S.J.E. acknowledges R35-GM136294 from the National Institutes of General Medical Sciences / NIH. D.D.N. would also like to acknowledge funding from the Searle Scholars Program sponsored by the Kinship Foundation, the Rose Hills Innovator Grant, the Beckman Young Investigator Award sponsored by the Arnold and Mabel Beckman Foundation and the Packard Fellowship in Science and Engineering sponsored by the David and Lucille Packard Foundation. D.D.N is a Chan-Zuckerberg Biohub – San Francisco Investigator. The funders had no role in the conceptualization and writing of this manuscript or the decision to submit the work for publication.

## Author contributions

D.G. contributed to conceptualization, data curation, formal analysis, methodology, and writing. K.C. contributed to data curation, formal analysis, methodology, and writing. S.J.E. contributed to conceptualization, data curation, formal analysis, supervision, funding acquisition, methodology, and writing. D.D.N contributed to conceptualization, data curation, formal analysis, supervision, funding acquisition, project administration, methodology, and writing.

## Competing interests

The authors declare no competing interests.
