## [Peer Review File · Nature Communications]

MmcA is an electron conduit that facilitates both intracellular and extracellular electron transport in *Methanosarcina acetivorans*

Editorial Note: Parts of this Peer Review File have been redacted as indicated to remove third party material where no permission to publish were obtainedReviewer #1 (Remarks to the Author):

The manuscript by Gupta et al. presents a comprehensive investigation into the biochemistry of a multiheme cytochrome, MmcA, known to be relevant for extracellular electron transfer in certain Methanosarcina species. The authors overexpressed MmcA under tetracycline induction, followed by purification and characterization, shedding light on its electron transfer capabilities.

One of the noteworthy findings of this study is the successful demonstration that MmcA is capable of reducing a methanophenazine analogue, 2-hydroxyphenazine. This observation strongly suggests that MmcA may play a vital role in the retrieval of extracellular electrons and their subsequent transfer to the methanophenazine pool. Furthermore, the study reveals that MmcA exhibits the ability to reduce AQDS and FeIII compounds, implying its potential involvement in extracellular electron transfer to reduce these constituents. However, the evidence regarding the effect of an mmcA deletion on FeIII metabolism appears somewhat weak.

The utilization of protein film voltammetry to characterize the redox-active range of this multiheme cytochrome provides compelling evidence of its capacity to transfer electrons to FeIII, AQDS, and MP.

The manuscript also delves into the relationship between MmcA and other multiheme cytochromes from Methanosarcinaceae, including MAGs of environmental species and MHCs from ANME. It is interesting, though somewhat loosely connected to the primary focus, and it remains unclear why the authors included this analysis in the manuscript.

Finally, there was one critical concern. The authors report that the mutant Δ mmcA used in their study exhibited a significant growth phenotype on typical methanogenic substrates. This mutant failed to utilize acetate and demonstrated impaired utilization of methylated compounds. Intriguingly, this contradicts the findings of Holmes (ref. 11 in this manuscript), who observed different behavior in their Δ mmcA mutant. This discrepancy raises suspicions that the gene deletion induced some effect (e.g. a frameshift), potentially resulting in issues at the translation and protein expression levels. Therefore, further clarification to address this inconsistency and to validate the observed phenotypic changes are necessary.

On the other hand, the Δ mmcA experiments may not be central to the manuscript's core message. Therefore, the authors might consider removing this section and exploring the comparison between their Δ mmcA mutant and Holmes' work at a later stage, thereby streamlining the manuscript's focus.

In conclusion, Gupta et al.'s manuscript provides valuable insights into the biochemistry of MmcA and its role in extracellular electron transfer. Nonetheless, the discrepancies in the mutant phenotypes require a more careful investigation, and the relevance of some of the analyses to the primary focus of the study should be clearly delineated.

Detailed comments:

Line 25: You demonstrated that MmcA donated electrons to a methanophenazine analogue, not directly to methanophenazine. Change to that.

Line 31: Remove "bioinformatically."

Line 33: "could act"

Lines 39-40: The division you present is inaccurate since all of them encode monoheme cytochromes.

Line 59: Remove "and many other important phenomena."

Lines 91-93: How is this possible, and how does it compare to the Lovley group's mutant?

Line 102: Can you clarify what you mean by "branch"?

Lines 117-118: Did it also restore the ability to use acetate?

Lines 115-123: This paragraph is not straightforward to follow; please rephrase for clarity.

Lines 153-155: This sentence suggests that the results with the *mmcA* k.o. by Holmes, which couldn't use AQDS, were insufficient to imply *MmcA*'s involvement in AQDS reduction. How should we then interpret your results regarding FeIII reduction by the *mmcA* k.o.?

Lines 161 and onward: This section should, in my opinion, be removed from this manuscript. It weakens the paper's core message. Extrapolating from a lab experiment using a mutant to predict outcomes in environmental samples is problematic. Not all *Methanosarcina* species have the *mmcA* gene, and making such assumptions is a significant stretch.

Lines 178-179: Instead of "overlapping envelope of reversible signals," explain in plain language suitable for a diverse readership, especially for a non-electrochemistry journal like *Nature Communications*.

Line 250: Does the structural analysis indicate any hydrophilicity? Can *MmcA* move between different membrane-bound proteins, possibly facilitating electron transfer?

Line 268: Reference 48 doesn't investigate iron reduction, but it is one of the first to discuss the multiheme cytochromes in *Methanosarcina* genomes and the absence of MHCs in others. While this reference appears to be relevant to your work, its placement beneath the current text is imprecise.

Lines 271-275: The conclusion needs a complete rewrite. For example, what do you mean by "tune the flow and path of electrons"? how did you measure that? The nice biochemical characterization of *MmcA* presented in this work deserves a more appropriate conclusion. The last sentence is unclear and should either be explained more clearly or removed. It's crucial to avoid suggesting potentially harmful environmental interventions like adding engineered strains or FeIII without addressing the associated issues that come with such practices (e.g. look at the iron fertilization experiments others did).

Reviewer #2 (Remarks to the Author):

Dinesh Gupta et al. investigate in molecular detail the biochemistry of the *Methanosarcina* c-type cytochrome *MmcA*. They purify the protein from *M. acetivorans*, analyze its heme groups and electrochemical properties. In addition, they provide conversion data of a previously published *deltaMmcA* mutant with the extracellular electron acceptors AQDS and Fe³⁺, in comparison to methanogenic growth conditions.

This study builds on many other papers that have already been published on *MmcA*, by this research group and others, e.g. Downing et al 2023, Holmes et al. 2022, Holmes et al. 2021, Holmes et al 2019. In this prior work, it has been demonstrated that *MmcA* is a c-type cytochrome involved in both electron transport during methanogenesis, as well as direct interspecies electron transfer and reduction of extracellular electron acceptors (AQDS, Fe(III)).

While I think the experiments in this paper are well conducted and scientifically sound, I miss an innovative research question or maybe the impact of the findings for the wider community. The

biochemical characterization provided here offers little new insights for a conceptual framework of, e.g., iron reduction by Methanosarcina. It is in line with previous findings and, in my opinion, would fit very well to a more biochemistry-oriented specialized journal, rather than Nature Comm.

Further comments:

1. The authors use Fe(II)/Fe(III) and Fe²⁺ and Fe³⁺ somewhat interchangeably. To my knowledge, Fe(II)/Fe(III) describe the oxidation states of the iron in minerals, such as hematite or magnetite, which would also be the natural electron acceptors (Fe(III)) for Methanosarcina. Fe(III) was not used in this study, and I question whether MmcA can actually reduce extracellular minerals as in their discussion the authors point out that Fe³⁺ would need to diffuse into the pseudoperiplasm – this can be done in the lab but is essentially unimportant in an environmental setting. I think the authors should report Fe³⁺ reduction rates and label it as such, to avoid the confusion with the minerals, or test minerals directly (e.g. in the form of iron nanoparticles).

2. While the authors argue that MmcA is important for Fe(III) reduction, the mmcA mutant retains 70% of the Fe³⁺ reducing activity. So one might argue that MmcA is only a minor constituent for this metabolic activity and other, more important, mechanisms are in place that need to be urgently investigated. I think if the authors were to combine a quest for those enzyme complexes with the results obtained here, the paper might have much more impact and wider implications for researchers in this field. Specifically, a double deletion with the archaellum would be interesting.

Minor comments:

L146 & 169: conditions/environments are anoxic, metabolism is anaerobic

L172-174: this seems presumptuous, it essentially says “we think mmcA is important under other conditions that no-one has ever looked at”. Either specify or delete.

Reviewer #3 (Remarks to the Author):

Summary:

The authors report deletion and phenotypic characterization of an mmcA deletion strain of *M. acetivorans*. MmcA is likely a new member of a class of membrane cytochrome involved in the electron transport chain of methanogens. They also overexpress the protein, enrich it in membrane preparations, and show chromatograms and cyclic voltammetry of protein preparations. The paper is presented well, but this reviewer has a few questions about methodology and figures that should be addressed before publication.

General questions and suggestions:

The electron transport chain is not as simple as described here, and one would not expect it to be. The oversimplified models runs the risk of misleading researchers who are new to the field. The authors must incorporate findings from several manuscripts in their model including:

<https://febs.onlinelibrary.wiley.com/doi/10.1111/febs.12031>;

<https://pubmed.ncbi.nlm.nih.gov/25232733/>; <https://onlinelibrary.wiley.com/doi/full/10.1111/j.1365-2958.2009.06990.x>, among others. Alternate routes of ferredoxin oxidation exist, and indeed the data presented here indicate MmcA is responsible for 25% of activity. Figures must be changed to reflect these findings and the manuscript needs to be edited throughout.

The authors suggest they were unable to measure reduction kinetics. In my experience the observations reported could be consistent with nonspecific chemistry. Perhaps appropriate controls were done but not shown in the paper or the supplementary information. Please address this question. The following are specific suggestions to improve the manuscript.

Specific questions and suggestions:

1. L30. Missing “is”.

2. L31. Delete "bioinformatically".
3. L34. Suggest rephrasing to "to potentially enable a variety of... beyond methanogenesis." To temper the speculation a bit.
4. SF1. Electron bifurcation should be shown, as this is critical for hydrogenotrophic methanogens. Oversimplifying this step gives the wrong impression to people new to the field. What are dotted lines?
5. SF2. Ion gradients are conventionally indicated by a Greek capital delta or Psi (depending on what you are indicating). Please adjust figures and legends throughout. What are dotted lines?
6. L56. Suggest changing to "If certain methanogens are indeed...". I would expect wide diversity in metabolic capabilities based on diversity on putative methanogen genomes discovered.
7. L64. Delete commas.
8. L77. Change to "...that certain methanogens are..."
9. L87. Suggest changing "inferred" to "concluded". One can always infer... whether or not the inference is correct is another issue.
10. L147-160. Instantaneous oxidation upon mixing is problematic. It should be possible to stoichiometrically titrate reductant and oxidant, which is easier and slower with proteins than with chemical redox reactions. Suggest additional controls may be needed to determine if there is an issue with the assay. These experiments and controls will allow you to calculate K_m for each substrate.
11. L172-174. Please remove this speculation or provide evidence of other acceptors.
12. F1 and L309. There is still quite a bit of contaminating membrane proteins in the MmcA prep, including some heme protein. To qualify as pure protein the prep should be >95%. Change "purified" to "enriched" throughout. Alternatively, please provide quantitative evidence of the purity of the MmcA prep used.
13. L325. I have several questions about the assays. A) How was the dithionite removed from the assays? Are you concerned excess dithionite could have interfered with the results? B) The way to show activity is to show stoichiometric oxidation of cytochrome with reduction of electron acceptor. Please show these data. C) As presented the reviewer cannot rule out that addition of components did not cause nonspecific oxidation of the hemes. Please include additional controls in the Supplementary information: vector only control prep, no dithionite, no substrate, etc) using mock or blank additions. SF9/SF10/SF11 are not sufficient for this purpose.
14. SF16. What evidence is there for the orientation of MmcA in the membrane as shown in this figure?

We thank the reviewers for their thoughtful and incisive comments. We have addressed the comments (see inline in red below) and have also performed some of the controls requested by reviewer # 3. We hope that the revised manuscript addresses the concerns raised by the reviewers suitably. Please note: the line numbers shown below correspond to the revised manuscript with track changes.

REVIEWER COMMENTS

Reviewer #1 (Remarks to the Author):

The manuscript by Gupta et al. presents a comprehensive investigation into the biochemistry of a multiheme cytochrome, MmcA, known to be relevant for extracellular electron transfer in certain Methanosarcina species. The authors overexpressed MmcA under tetracycline induction, followed by purification and characterization, shedding light on its electron transfer capabilities.

One of the noteworthy findings of this study is the successful demonstration that MmcA is capable of reducing a methanophenazine analogue, 2-hydroxyphenazine. This observation strongly suggests that MmcA may play a vital role in the retrieval of extracellular electrons and their subsequent transfer to the methanophenazine pool. Furthermore, the study reveals that MmcA exhibits the ability to reduce AQDS and FeIII compounds, implying its potential involvement in extracellular electron transfer to reduce these constituents. However, the evidence regarding the effect of an mmcA deletion on FeIII metabolism appears somewhat weak.

The utilization of protein film voltammetry to characterize the redox-active range of this multiheme cytochrome provides compelling evidence of its capacity to transfer electrons to FeIII, AQDS, and MP.

The manuscript also delves into the relationship between MmcA and other multiheme cytochromes from Methanosarcinaceae, including MAGs of environmental species and MHCs from ANME. It is interesting, though somewhat loosely connected to the primary focus, and it remains unclear why the authors included this analysis in the manuscript.

Finally, there was one critical concern. The authors report that the mutant Δ mmcA used in their study exhibited a significant growth phenotype on typical methanogenic substrates. This mutant failed to utilize acetate and demonstrated impaired utilization of methylated compounds. Intriguingly, this contradicts the findings of Holmes (ref. 11 in this manuscript), who observed different behavior in their Δ mmcA mutant. This discrepancy raises suspicions that the gene deletion induced some effect (e.g. a frameshift), potentially resulting in issues at the translation and protein expression levels. Therefore, further clarification to address this inconsistency and to validate the observed phenotypic changes are necessary.

We thank the reviewer for this very valid concern regarding the difference in phenotypes that have been reported for the Δ mmcA mutant by our group and by the Lovley group. We have performed whole genome sequencing of our Δ mmcA mutant. See Supplementary Table 1 from Downing et al. 2023 (<https://onlinelibrary.wiley.com/doi/full/10.1111/mmi.15029>) below and the raw reads from Illumina sequencing of this mutant can be found online ([https://www.ncbi.nlm.nih.gov/sra/SRX19023896\[accn\]](https://www.ncbi.nlm.nih.gov/sra/SRX19023896[accn])).

Supplementary Table 1: List of mutations in CRISPR-edited mutant strain DDN009 containing a Δ *mmcA* in-frame deletion mutation

Position	Mutation	Notes
487691	C added	Present in Parent (WWM60)
774965	Δ1389 bp	ΔmmcA (only present in DDN009)
836182-836771	Δhpt::PmcrB-tetR	Present in Parent (WWM60)
941168	A5->6	Present in Parent (WWM60)
1314120	Δ 1bp	Present in Parent (WWM60)
2086881-82	TC->CT	Present in Parent (WWM60)
2086886	G->T	Present in Parent (WWM60)
2534543	C added	Present in Parent (WWM60)
2836646	A->G (F64L)	Present in Parent (WWM60)
2867059	T->G (H313Q)	Present in Parent (WWM60)
3433201	Δ 1bp	Present in Parent (WWM60)
4295452	Δ 1bp	Present in Parent (WWM60)
4874567	Δ 1bp	Present in Parent (WWM60)
4945345	C added	Present in Parent (WWM60)
5078585	G->A (M1M)	Present in Parent (WWM60)

We were unable to find any frameshift mutations or loss of function mutations in genes involved in methanogenesis (on acetate or methylated compounds) in our Δ *mmcA* mutant.

In addition, we have observed that complementing *mmcA* back on a plasmid in the Δ *mmcA* mutant rescues the growth defect on methylated compounds; see Figure 2e from Downing et al. 2023 (<https://onlinelibrary.wiley.com/doi/full/10.1111/mmi.15029>) below.

Based on these observations, we are confident that the phenotype we have reported in this study and others is truly due to the deletion of the *mmcA* locus. In addition, we have incubated our Δ *mmcA* mutant in media with acetate for over a year to select for suppressors but have not obtained any growth for this mutant. We have requested the Δ *mmcA* mutant from the Lovley lab for sequence validation but have been unable to obtain the strain so we cannot speak to the validity of their strain but are certain that the Δ *mmcA* mutant we have generated does not contain any second-site mutation

that is obscuring the phenotype observed. We hope this information alleviates the reviewer's concerns.

On the other hand, the Δ mmcA experiments may not be central to the manuscript's core message. Therefore, the authors might consider removing this section and exploring the comparison between their Δ mmcA mutant and Holmes' work at a later stage, thereby streamlining the manuscript's focus.

We refer the reviewer to the extensive work we have done (see note above) to validate the genotype and phenotype of our Δ mmcA mutant. We are confident in the genotype and phenotype of our mutant and would politely request that we keep the section of the manuscript as is given the time and effort we have spent on validation.

In conclusion, Gupta et al.'s manuscript provides valuable insights into the biochemistry of MmcA and its role in extracellular electron transfer. Nonetheless, the discrepancies in the mutant phenotypes require a more careful investigation, and the relevance of some of the analyses to the primary focus of the study should be clearly delineated.

Detailed comments:

Line 25: You demonstrated that MmcA donated electrons to a methanophenazine analogue, not directly to methanophenazine. Change to that.

We have made the change requested by the reviewer. Please see lines 24-26 in the revised manuscript.

Line 31: Remove "bioinformatically."

We have removed the word "bioinformatically" per the reviewer's request. Please see line 32 in the revised manuscript.

Line 33: "could act"

We have replaced 'acts' with 'might act' per the reviewer's request. Please see line 34 in the revised manuscript.

Lines 39-40: The division you present is inaccurate since all of them encode monoheme cytochromes.

We respectfully disagree with the reviewer. Most methanogens lack any cytochromes; a small subset, mostly members of the *Methanosarcinales*, contain cytochromes. Within the *Methanosarcinales*, many strains only encode *b*-type cytochromes (like *Methanosarcina barkeri*) and some others (like *Methanosarcina acetivorans*) also encode *c*-type cytochromes. This is a well-established classification in the field and we refer the reviewer to this seminal paper that describes the basis for this classification scheme: <https://www.nature.com/articles/nrmicro1931>

Line 59: Remove "and many other important phenomena."

We have removed this phrase per the reviewer's request. Please see line 60 in the revised manuscript.

Lines 91-93: How is this possible, and how does it compare to the Lovley group's mutant?

We refer the reviewer to the extensive validation we have done to verify the phenotype and genotype of the $\Delta mmcA$ mutant (see note above). We have requested the mutant from the Lovley group repeatedly but have not heard from them yet. They have not sequenced the mutant they have generated (or this information is not publicly available) so we cannot speak of the genotype of their strain.

Line 102: Can you clarify what you mean by "branch"?

We have rephrased the sentence per the reviewer's request to 'Overall, MmcA is a versatile electron carrier in the ETC of methanogens and can facilitate energy conservation by methanogenesis or iron respiration depending on the availability of oxidized iron species in the environment.' See lines 104-105 in the revised manuscript.

Lines 117-118: Did it also restore the ability to use acetate?

We have not tested the growth of the *mmcA* complementation on acetate but, given that it restores growth on methylated compounds, we expect that growth will be restored on acetate too. The experiment to test complementation on acetate is more time consuming and will take >2 months and not necessary hence was not conducted.

Lines 115-123: This paragraph is not straightforward to follow; please rephrase for clarity.

Per the reviewer's request, we have rephrased it for clarity. Please see lines 118-126 in the revised manuscript.

Lines 153-155: This sentence suggests that the results with the *mmcA* k.o. by Holmes, which couldn't use AQDS, were insufficient to imply MmcA's involvement in AQDS reduction. How should we then interpret your results regarding FeIII reduction by the *mmcA* k.o.?

We thank the reviewer for their insightful comments. Based on the conclusions of the Holmes et al. 2019 study, a mutant of *M. acetivorans* lacking *mmcA* cannot reduce AQDS. These genetic data suggest that **MmcA plays a role in AQDS reduction but does not show that MmcA directly mediates this process.** Our work with purified protein shows that MmcA does, indeed, directly mediate this process. Even though MmcA can reduce AQDS/Fe³⁺, our data clearly show that *M. acetivorans* has alternate mechanisms for these reactions. These could be other α -type cytochromes encoded in the genome or cytochrome independent mechanisms as have been proposed previously (such as this notable paper in this area: <https://www.nature.com/articles/s41598-019-57206-z>). We hope that our lab and others in the field will uncover the identity of these mechanisms soon but that is beyond the scope of this work. In this work, we are outlining the various redox reactions that MmcA can catalyze in methanogens and the implications of those reactions on the physiology of the organism.

We would also like to point out that many microorganisms that perform EET like *Geobacter sp.* or *Shewanella sp.* have redundant pathways for metal reduction. Deletion of any one pathway often has

no effect on AQDS/metal reduction. For instance, see figure 1 from <https://journals.asm.org/doi/epub/10.1128/aem.02250-09> below where a single omcS/E/B/Z deletion completely obliterates the capacity of *Geobacter* to mediate AQDS/humic acid reduction. Each single mutant barely any defect on metal reduction. Hence it is not surprising that that the $\Delta mmcA$ mutant only has a 30% defect in iron reduction rate.

Lines 161 and onward: This section should, in my opinion, be removed from this manuscript. It weakens the paper's core message. Extrapolating from a lab experiment using a mutant to predict outcomes in environmental samples is problematic. Not all *Methanosarcina* species have the *mmcA* gene, and making such assumptions is a significant stretch.

We thank the reviewer for this comment but would like to appeal to them that this section provides some perspective on the link between data generated in the lab and its environmental implications. Microbial geneticists often disregard mutations that have a small defect in lab as not being important, but we wanted to emphasize that small defects can compound over the course of multiple generations in the environment to have a large impact. While we know that not all *Methanosarcina* have *MmcA*, for those that do, it is clearly important in its environmental context. We hope the reviewer sees the validity in these statements but are open to making any modifications if needed.

Lines 178-179: Instead of "overlapping envelope of reversible signals," explain in plain language suitable for a diverse readership, especially for a non-electrochemistry journal like *Nature Communications*.

This is a great point. We have rephrased this sentence to make it accessible to a reader who might not be as well versed with electrochemistry. Please see lines 188-189 in the revised manuscript.

Line 250: Does the structural analysis indicate any hydrophilicity? Can *MmcA* move between different membrane-bound proteins, possibly facilitating electron transfer?

There are no predicted transmembrane domains MmcA so it is unlikely to be a membrane integral or membrane bound protein and likely associates with other membrane-bound proteins. That said, the reviewer brings up a great point. In our previous work, we observed that expressing MmcA without the rest of the Rnf complex does seem to provide a growth advantage (see Downing et al. 2023: <https://onlinelibrary.wiley.com/doi/full/10.1111/mmi.15029>, Figure 4c; pasted below). These data indicate that MmcA might be able to move between different membrane proteins, but we don't have any conclusive evidence for how and what other proteins it might interact with. We have conducted co-immunoprecipitation experiments with FLAG-tagged MmcA and have not been able to reliably identify any candidates so far, likely because these interactions are transient or unstable. We are hoping to use crosslinkers to overcome these experimental hurdles soon.

Line 268: Reference 48 doesn't investigate iron reduction, but it is one of the first to discuss the multiheme cytochromes in *Methanosarcina* genomes and the absence of MHCs in others. While this reference appears to be relevant to your work, its placement beneath the current text is imprecise.

We thank the reviewer for this insightful comment. We have modified the sentence to ‘Alternately, it is entirely plausible that alternate pathway(s) for iron respiration are MHC-independent as members of the *Methanosarcinales* lacking *cyt c* have also been shown to reduce iron or participate in direct interspecies electron transfer (DIET). Please see lines 278-279 in the revised manuscript.

Lines 271-275: The conclusion needs a complete rewrite. For example, what do you mean by "tune the flow and path of electrons"? how did you measure that? The nice biochemical characterization of MmcA presented in this work deserves a more appropriate conclusion. The last sentence is unclear and should either be explained more clearly or removed. It's crucial to avoid suggesting potentially harmful environmental interventions like adding engineered strains or FeIII without addressing the associated issues that come with such practices (e.g. look at the iron fertilization experiments others did).

Yes, this is a very valid point. To address the reviewer's concern, we have rewritten the conclusions section and removed any sentences that might allude to dangerous practices like Fe(III) amendment in natural environments altogether. Please see lines 282-290 in the revised manuscript.

Reviewer #2 (Remarks to the Author):

Dinesh Gupta et al. investigate in molecular detail the biochemistry of the Methanosarcina c-type cytochrome MmcA. They purify the protein from *M. acetivorans*, analyze its heme groups and electrochemical properties. In addition, they provide conversion data of a previously published Δ MmcA mutant with the extracellular electron acceptors AQDS and Fe³⁺, in comparison to methanogenic growth conditions.

This study builds on many other papers that have already been published on MmcA, by this research group and others, e.g. Downing et al 2023, Holmes et al. 2022, Holmes et al. 2021, Holmes et al 2019. In this prior work, it has been demonstrated that MmcA is a c-type cytochrome involved in both electron transport during methanogenesis, as well as direct interspecies electron transfer and reduction of extracellular electron acceptors (AQDS, Fe(III)).

While I think the experiments in this paper are well conducted and scientifically sound, I miss an innovative research question or maybe the impact of the findings for the wider community. The biochemical characterization provided here offers little new insights for a conceptual framework of, e.g., iron reduction by Methanosarcina. It is in line with previous findings and, in my opinion, would fit very well to a more biochemistry-oriented specialized journal, rather than Nature Comm.

We thank the reviewer for their insightful comments but respectfully disagree with their statement on what has been previously demonstrated. Genetic analyses by our group and others have shown that the Δ *mmcA* mutant has a growth defect during methanogenesis and AQDS reduction but these data, in and of themselves, do not indicate what MmcA does.

To assess the biochemical function of MmcA, one needs pure protein, and obtaining pure protein from a methanogen is especially challenging and requires, both, genetic and biochemical expertise. **This is why, to the best of our knowledge, there have been no studies that have purified a c-type cytochrome from methanogens and studied its biochemical and electrochemical properties.** We are the first group to purify MmcA and test all the hypotheses re: its function, hence this work truly does represent a quantum leap in our understanding of this important component of the ETC in methanogens. **Our biochemical and electrochemical studies support some of the previous hypotheses re: the role of MmcA, for sure, but that does not undercut the novelty of our work. It provides conclusive and robust evidence for an important phenomenon when there was none in the past.**

We hope the reviewer also appreciates the technical breakthrough this work represents in and of itself and by paving the way for many such studies in the future with other important cytochromes involved in extracellular electron transfer in methanogens and anaerobic methane oxidizing archaea (ANME). Given the broad relevance of our techniques and the outcomes of this study to a diverse community of researchers studying methanogens and ANMEs from different perspectives, we truly believe that this work is well within the scope of an interdisciplinary journal like Nature Communications rather than a specialized biochemistry journal. Indeed, if this was just another c-type cytochrome being studied in a well-established model system like *Geobacter* or *Shewanella*, the

paper would be more appropriate for a biochemistry oriented specialized journal but that is not the case here that we hope to convince the reviewer of.

Further comments:

1. The authors use Fe(II)/Fe(III) and Fe²⁺ and Fe³⁺ somewhat interchangeably. To my knowledge, Fe(II)/Fe(III) describe the oxidation states of the iron in minerals, such as hematite or magnetite, which would also be the natural electron acceptors (Fe(III)) for *Methanosarcina*. Fe(III) was not used in this study, and I question whether MmcA can actually reduce extracellular minerals as in their discussion the authors point out that Fe³⁺ would need to diffuse into the pseudoperiplasm – this can be done in the lab but is essentially unimportant in an environmental setting. I think the authors should report Fe³⁺ reduction rates and label it as such, to avoid the confusion with the minerals, or test minerals directly (e.g. in the form of iron nanoparticles).

We thank the reviewer for their insightful feedback and have reported all iron reduction rates as Fe³⁺ reduction rates and have used the nomenclature they have prescribed throughout the text. While it remains unclear if *Methanosarcina* can reduce Fe(III) in minerals, it is clear that any soluble Fe³⁺ can be used as an electron acceptor through the action of MmcA. As to the environmental relevance of this phenomenon, we would like to point out that the amount of soluble Fe has been on the rise due to anthropogenic activity of late (see <https://agupubs.onlinelibrary.wiley.com/doi/full/10.1029/2008GB003440> for instance), which makes our work especially relevant in the context of climate change.

2. While the authors argue that MmcA is important for Fe(III) reduction, the *mmcA* mutant retains 70% of the Fe³⁺ reducing activity. So one might argue that MmcA is only a minor constituent for this metabolic activity and other, more important, mechanisms are in place that need to be urgently investigated. I think if the authors were to combine a quest for those enzyme complexes with the results obtained here, the paper might have much more impact and wider implications for researchers in this field. Specifically, a double deletion with the archaellum would be interesting.

We thank the reviewer for their comment. We have three different reasons why we think MmcA is still important for Fe³⁺ reduction in methanogens, even though a *mmcA* deletion retains 70% activity as noted below.

First, based on previous studies with model organisms like *Geobacter sp.* or *Shewanella sp.*, microbes often encode redundant pathways for AQDS/humic acids/metal reduction. Deletion of any one pathway often has no effect on AQDS/humic acids/metal reduction. For instance, see figure 1 from <https://journals.asm.org/doi/epub/10.1128/aem.02250-09> where a single *omcS/E/B/Z* deletion has little to no impact on the capacity of *Geobacter* to mediate AQDS/humic acid reduction. If these genetic data were used to interpret that OmcS/E/B/Z are not important for EET, then the field of electromicrobiology would have suffered tremendously. Instead, follow up work on each of these proteins have enriched the field tremendously.

Based on work done in *Geobacter sp.* or *Shewanella sp.*, it is likely that electroactive methanogen like *M. acetivorans* might also alternate mechanisms for Fe³⁺ reduction besides MmcA. **The presence of alternate mechanisms does not undercut the importance of MmcA in this process.** Since research on EET in methanogens is relatively new compared to the decades of work done in *Geobacter* or *Shewanella* it will take a while for our lab and others to figure out all the pathways

involved. Regardless, redundancy is a feature not a bug in *M. acetivorans* and should not discount the value or importance of MmcA for future work in the field.

Second, we would like to point out that even though there is one paper (Holmes et al mBio 2021: <https://journals.asm.org/doi/full/10.1128/mbio.02344-21>) that alludes to the importance of archaella in electron transfer in *M. acetivorans*, many groups have previously reported that even though the genes for making an archaella are encoded in the genome of *M. acetivorans*, it has not been detected by standard assays or electron microscopy (for example, see excerpt from Galagan et al. 2002 <https://genome.cshlp.org/content/12/4/532.short> below)

Despite the presence of flagellin and chemotaxis genes, we were unable to detect processing of preflagellin in a preflagellin peptidase assay using *Methanococcus voltae* preflagellin and membranes obtained from *M. acetivorans* growing as single cells (Correia and Jarrell 2000). Thus, it seems likely that flagella and chemotaxis genes are expressed in specific environmental conditions not previously created in culture (as has been seen for other organisms [Faguy et al. 1993; Mukhopadhyay et al. 2000]), possibly in conjunction with changes in morphology.

We believe that a thorough study needs to be performed to identify the conditions under which the archaellum is expressed in *M. acetivorans* and its role, if at all, in electron transfer. In addition, we think a broader and unbiased approach, such as high-throughput assays to measure Fe⁺³-reduction in a transposon mutant library of $\Delta mmcA$ would be a more productive way to identify alternate

pathways (as described in the text; see lines 279-281). Furthermore, any work involving mutants of the archaeum is beyond the scope of our current study and would take us months/years to conduct at the level of rigor we feel comfortable reporting.

Finally, as the reviewer points out a 30% defect in Fe^{3+} reduction rates might be considered minor under lab conditions but, when this defect is compounded over generations, it can have a severe impact on the fitness of the organism. A mutant with a 30% defect in growth rate (or energy conservation) would be outnumbered by at least 100-fold in 30 generations and this ratio will grow exponentially with every subsequent passage.

We hope that with these three distinct arguments we can convince the reviewer that a 30% defect in iron reduction rates for the ΔmmcA mutant is expected (due to redundancy) but still extremely important from an ecological standpoint.

Minor comments:

L146 & 169: conditions/environments are anoxic, metabolism is anaerobic

We thank the reviewer for this insightful comment and have made the changes requested. Please see lines 151 and 179 in the revised manuscript.

L172-174: this seems presumptuous, it essentially says “we think *mmcA* is important under other conditions that no-one has ever looked at”. Either specify or delete.

We have removed this sentence per the reviewer’s request. Please see lines 182-184 in the revised manuscript.

Reviewer #3 (Remarks to the Author):

Summary:

The authors report deletion and phenotypic characterization of an *mmcA* deletion strain of *M. acetivorans*. *MmcA* is likely a new member of a class of membrane cytochrome involved in the electron transport chain of methanogens. They also overexpress the protein, enrich it in membrane preparations, and show chromatograms and cyclic voltammetry of protein preparations. The paper is presented well, but this reviewer has a few questions about methodology and figures that should be addressed before publication.

We thank the reviewer for their balanced feedback. We have addressed the concerns of the reviewer by clarifying the text and performing appropriate controls as requested.

General questions and suggestions:

The electron transport chain is not as simple as described here, and one would not expect it to be. The oversimplified model runs the risk of misleading researchers who are new to the field. The authors must incorporate findings from several manuscripts in their model including: <https://febs.onlinelibrary.wiley.com/doi/10.1111/febs.12031>; <https://pubmed.ncbi.nlm.nih.gov/25232733/>; <https://onlinelibrary.wiley.com/doi/full/10.1111/j.1365-2958.2009.06990.x>, among others. Alternate routes of ferredoxin oxidation exist, and indeed the data presented here indicate *MmcA* is responsible for 25% of activity. Figures must be changed to reflect these findings and the manuscript needs to be edited throughout.

The reviewer is absolutely correct. The ETC of methanogens is nowhere near as simple as portrayed in our paper and there are many unknown elements that are yet to be discovered. That said, the goal of this paper is to study the role of MmcA in the Rnf complex hence we have centered our view of the ETC on Rnf. That said, we do not want to mislead anyone and have changed Supplementary Figures 1 & 2 substantially per the reviewer's request. We also apologize for our oversight and have added references to the seminal papers in the field per the reviewer's request (Please see the figure legend of supplementary figures 1 and 2 in the supplement).

The authors suggest they were unable to measure reduction kinetics. In my experience the observations reported could be consistent with nonspecific chemistry. Perhaps appropriate controls were done but not shown in the paper or the supplementary information. Please address this question. The following are specific suggestions to improve the manuscript.

This is a great point and we have made modifications to the manuscript and conducted additional experiments to address this extremely valid concern.

First, the redox reactions catalyzed by cytochromes with electron acceptors like AQDS or Fe^{3+} are very fast and proceed to completion in less than 10 milliseconds. See Figure 2 from: <https://www.sciencedirect.com/science/article/pii/S0005272814000516> and other relevant studies like <https://www.sciencedirect.com/science/article/pii/S0005272811000053> or <https://www.sciencedirect.com/science/article/pii/S0005272818301221>.

These rates need to be measured using stopped-flow apparatus that we do not have access to. To address the reviewer's concern, we have modified the text in the manuscript as shown below.

*“All three electron acceptors oxidized MmCA rapidly as confirmed by the change in the spectral profile of the protein (Figure 3; Supplementary Figures 10 and 11). In contrast, a control experiment with Zn^{+2} as the sole electron acceptor did not change the redox state of dithionite-reduced MmCA (Supplementary Figures 12 and 13). While we were unable to measure the reaction kinetics using our experimental setup, these data confirm the interaction between MmCA and AQDS or Fe^{3+} and validate its important role during anaerobic respiration in *M. acetivorans*.”*

Please see lines 164-170 of the revised manuscript.

Specific questions and suggestions:

1. L30. Missing “is”.

We have corrected this typo and thank the reviewer for bringing it to our attention. Please see line 30 of the revised manuscript.

2. L31. Delete “bioinformatically”.

We have deleted this word. Please see line 32 of the revised manuscript.

3. L34. Suggest rephrasing to “to potentially enable a variety of... beyond methanogenesis.” To temper the speculation a bit.

We have rephrased the text per the reviewer’s suggestion. Please see lines 34-35 of the revised manuscript.

4. SF1. Electron bifurcation should be shown, as this is critical for hydrogenotrophic methanogens. Oversimplifying this step gives the wrong impression to people new to the field. What are dotted lines?

Indeed, we have changed this figure to show electron bifurcation. We have provided more details re: individual reactions as requested by the reviewer and to avoid any confusion.

5. SF2. Ion gradients are conventionally indicated by a Greek capital delta or Psi (depending on what you are indicating). Please adjust figures and legends throughout. What are dotted lines?

We thank the reviewer for their feedback. In this case, we aren’t necessarily showing ion gradients rather showing the direction of ion translocation by the various complexes. We have clarified this important detail in the revised figure legend. We have also provided more details in terms of the enzymes that the dotted lines represent and the alternate routes for ferredoxin re-oxidation (such as HdrABC) based on previous work too.

6. L56. Suggest changing to “If certain methanogens are indeed...”. I would expect wide diversity in metabolic capabilities based on diversity on putative methanogen genomes discovered.

We have revised the text based on the reviewer’s suggestion. Please see line 57 of the revised manuscript.

7. L64. Delete commas.

We have revised the text based on the reviewer’s suggestion. Please see line 65 of the revised manuscript.

8. L77. Change to “...that certain methanogens are...”

We have revised the text based on the reviewer’s suggestion. Please see line 79 of the revised manuscript.

9. L87. Suggest changing “inferred” to “concluded”. One can always infer... whether or not the inference is correct is another issue.

We have revised the text based on the reviewer’s suggestion. Please see lines 88-89 of the revised manuscript.

10. L147-160. Instantaneous oxidation upon mixing is problematic. It should be possible to stoichiometrically titrate reductant and oxidant, which is easier and slower with proteins than with chemical redox reactions. Suggest additional controls may be needed to determine if there is an issue with the assay. These experiments and controls will allow you to calculate K_m for each substrate.

We refer the reviewer to our note above re: the rates of redox reactions catalyzed by c-type cytochromes and how our current experimental setup is not capable of stopped flow experiments hence cannot measure these fast rates. We have qualified our results in the text appropriately. Please see lines 164-170 of the revised manuscript.

We have also conducted controls reactions where we have incubated pre-reduced MmcA with Zn^{2+} (Please see supplementary Figures 12 and 13, and lines 166-167 in the revised manuscript). Although 100 μM of AQDS or Fe^{3+} could completely oxidize the dithionite-reduced MmcA (Figure 3), addition of Zn^{2+} (100-400 μM) did not oxidize the reduced-MmcA (Please see Supplementary Figure 12 and 13 in the manuscript). These results are also supported based on the redox range of the heme groups in MmcA and the redox potential of the electron acceptors. For example, the reduction AQDS or Fe^{3+} by MmcA is thermodynamically favorable whereas the reduction of Zn^{2+} by MmcA is thermodynamically unfavorable. Accordingly, we observe that reduced-MmcA gets oxidized by AQDS or Fe^{3+} but not by Zn^{2+} . These results show that the readout of our assays is due to a reaction between MmcA and the electron acceptor and not due to any chemical reactions occurring in the buffer or due to excess amount of sodium dithionite in the reaction.

11. L172-174. Please remove this speculation or provide evidence of other acceptors.

We have removed this speculative statement per the reviewer's request. Please see lines 182-184 of the revised manuscript.

12. F1 and L309. There is still quite a bit of contaminating membrane proteins in the MmcA prep, including some heme protein. To qualify as pure protein the prep should be >95%. Change "purified" to "enriched" throughout. Alternatively, please provide quantitative evidence of the purity of the MmcA prep used.

We have changed purified to enriched per the reviewer's request. That said, we would like to note that even though there are additional bands on our Coomassie gel, none of them light up with a heme stain i.e., none of them are c-type cytochromes. Regardless, we are happy to change purified to enriched and exercise appropriate caution as recommended by the reviewer.

13. L325. I have several questions about the assays. A) How was the dithionite removed from the assays? Are you concerned excess dithionite could have interfered with the results? B) The way to show activity is to show stoichiometric oxidation of cytochrome with reduction of electron acceptor. Please show these data. C) As presented the reviewer cannot rule out that addition of components did not cause nonspecific oxidation of the hemes. Please include additional controls in the Supplementary information: vector only control prep, no dithionite, no substrate, etc) using mock or blank additions. SF9/SF10/SF11 are not sufficient for this purpose.

We thank the reviewer for their valid concerns here.

- A) We did not remove dithionite and do not expect it to interfere with the results as the goal of our experiments is to test the ability of electron acceptors to oxidize MmcA. Under the conditions tested, dithionite would act as an electron donor not as an electron acceptor. While the reviewer is right that the leftover (excess of) dithionite would reduce some of these oxidized-MmcA back to its reduced form. To account for this, we added AQDS and Fe^{3+} in excess (~10 times more compared to sodium dithionite) in our experiments. Also, we did not observe any oxidation of MmcA due to excessive dithionite in our control experiment with Zn^{2+} .
- B) Please see our note above re: reaction rates and how we are incapable of measuring reaction kinetics. Given the limitations of our experimental setup, we will not be able to distinguish between the experimental outcomes for different ratios of the MHC and the electron acceptor. We have now clearly stated the limitations of this experiment in the text (see lines 167-170 in the revised manuscript) and do not anticipate that the absence of these data undercut our major result which is to show that MmcA can interact with and donate electrons to extracellular electron acceptors like AQDS and Fe^{3+} .
- C) We have performed control reactions with an additional electron acceptor (Zn^{2+}). Electron transfer from MmcA to Zn^{2+} is thermodynamically unfavorable, which is consistent with our experimental outcome (see Supplementary Figure 12 and 13). These data clearly indicate that our reaction buffer is not interfering with our assay. We hope that these data convince the reviewer of the validity of the assays used in our study.

14. SF16. What evidence is there for the orientation of MmcA in the membrane as shown in this figure?

We thank the reviewer for this comment and clarified the figure legend to address their question. The structure in this figure is based on the AlphaFold model of MmcA and we are showing the orientation of MmcA that fits with the orientation of the other subunits of the Rnf complex as described recently in <https://www.nature.com/articles/s41467-022-34007-z>. We also performed AlphaFold modeling to predict the protein-protein interaction between MmcA and a few components of Rnf-complex like RnfG (it has transmembrane domain as well as periplasmic exposed part), and RnfX (an integral membrane protein), which are also consistent with the orientation shown in the figure. Finally, even though the model shows MmcA in one orientation, we can change its orientation and that would still not impact the organization of the heme groups within the protein. We have clarified these details in the figure legends.

Reviewer #2 (Remarks to the Author):

The authors have addressed all comments I raised on the minor points, and have argued on the innovative and groundbreaking nature of their results, which I raised as a major concern. The minor concerns have been addressed appropriately.

For the major concerns, however, I am unable to change my opinion. It does not question the validity of the results of the authors, but rather the impact, novelty and groundbreaking nature of their results. The authors argue that the biochemical function was previously not known, whereas in my opinion other experiments have already suggested this function (documented in numerous papers mentioned in my first review, e.g. by deletion mutant studies). I do not see the confirmation of previous experiments via a different experimental approach as groundbreaking, more as affirmatory - which is fine, but in my view does not warrant publication in Nature communications. They argue that this article is the first to investigate isolated c-type cytochromes in methanogenic archaea. Whereas this is undoubtedly true, I think the innovation doesn't come from doing something for the first time, but by the insights gained from this - and I couldn't find groundbreaking new insights resulting from the author's experiments. In this paper, results are either affirmatory of previous experiments or descriptive, which I think is not groundbreaking. The authors further argue that this opens the way to study ANME cytochromes. This may or may not be the case, so the innovation then would come from actually studying those cytochromes, not suggesting that it is possible to do so.

In essence, I'm curious how the editor will decide whether such results can be seen as highly innovative and groundbreaking, to be publishable in Nature Comm. In my opinion, the impact of the results of the authors are not warranting this.

Reviewer #3 (Remarks to the Author):

The authors have addressed my concerns.

Reviewer #4 (Remarks to the Author):

The manuscript by Gupta et al. provides a thorough characterization of the multiheme cytochrome of *Methanosarcina MmcA*. This protein is believed to play a central role in reduction of methanophenazine during methanogenesis from methyl compounds and acetate. The biochemical work is well done. As this paper is the first description of this protein from a methanogen, it is of some importance.

Overall the paper is well written and clear. A few points that might be reconsidered are as follows.

In the introduction, methanogens with and without cytochromes are contrasted. Methanogens 'with cytochromes' is inaccurate since many of these species also use H₂/CO₂ to make methane and also have many of the enzymes used by the 'methanogens without cytochromes'. A clearer and more accurate distinction would be hydrogenotrophic vs methylotrophic/acetotrophic pathways of methanogenesis. Likewise, the figure S2 fails to include the mtr which is critical to methanogenesis from acetate. Since it is supposed to show the pathway of methanogenesis, the figure would be clearer if mtr and mcr were included. This distinction is important because mtr is critical to the bioenergetics of acetotrophic growth.

Lines 173-178. The argument for the physiological role of MmcA in iron reduction is not necessary and detracts from the overall discussion. First, the obvious interpretation of Fig. 4 is that MmcA plays a minor or no role in iron anaerobic respiration is ignored. A 30% reduction in activity could easily have resulted from pleiotropic effects. Second, this experiment cannot rule out redundancy in the pathway, so the results of a single mutation are difficult to interpret. Third, reduction was measured and not

anaerobic respiration, so it is also possible that the reduction was just a side reaction. Lastly, given the extensive washing that was performed on the cells, it would be nice to compare the rates of iron reduction with those measured during growth with iron, which are available in the literature. This comparison would ensure that the major activity was being measured.

Lines 222-225. The claim that the manuscript demonstrates iron-dependent respiration is not supported by the evidence. Only iron reduction was measured. Evidence was not presented for generation of a proton motive force or ATP, which is required to claim respiration. The authors could delete this very speculative paragraph without detriment to the value of the paper.

Lines 241-250. This paragraph seems to be premature and could be deleted.

Line 203. A word seems to be missing in this sentence.

Reviewer #5 (Remarks to the Author):

As an adjudicating reviewer, I have carefully considered how novel and rigorous this manuscript in the context of the three previous reviews. Overall, the manuscript represents a major advance as the first biochemical characterisation of a c-type cytochrome in methanogens, benefiting from strong activity studies and electrochemical analysis. I agree that, on the basis of the mild effects observed in the mutant data, the physiological role of this enzyme remains an open question. But I am personally satisfied that the manuscript makes a sufficient independent advance irrespectively and think the authors have responded appropriately to the reviewers.

We thank the reviewers for their thoughtful comments. We have addressed each of the comments below (in red). Please note that the line numbers correspond to the revised manuscript.

Reviewer #4 (Remarks to the Author):

The manuscript by Gupta et al. provides a thorough characterization of the multiheme cytochrome of *Methanosarcina MmcA*. This protein is believed to play a central role in reduction of methanophenazine during methanogenesis from methyl compounds and acetate. The biochemical work is well done. As this paper is the first description of this protein from a methanogen, it is of some importance.

Overall the paper is well written and clear. A few points that might be reconsidered are as follows.

We thank the reviewer for their insightful feedback. We have addressed the concerns of the reviewer by removing and clarifying the text in the manuscript.

In the introduction, methanogens with and without cytochromes are contrasted. Methanogens 'with cytochromes' is inaccurate since many of these species also use H₂/CO₂ to make methane and also have many of the enzymes used by the 'methanogens without cytochromes'. A clearer and more accurate distinction would be hydrogenotrophic vs methylotrophic/acetotrophic pathways of methanogenesis. Likewise, the figure S2 fails to include the mtr which is critical to methanogenesis from acetate. Since it is supposed to show the pathway of methanogenesis, the figure would be clearer if mtr and mcr were included. This distinction is important because mtr is critical to the bioenergetics of acetotrophic growth.

We respectfully disagree with the reviewer. The distinction between methanogens with and without cytochromes is well-established and has been used in the field for more than a decade. We refer the reviewer to this seminal paper that describes the basis for this classification scheme:

<https://www.nature.com/articles/nrmicro1931>

The reviewer suggests that we categorize methanogens based on their substrate range. This scheme would be misleading as methanogens with and without cytochromes can grow on H₂+CO₂ but the underlying energy conservation pathways are completely distinct. The former i.e. methanogens without cytochromes use the Mtr complex to generate a sodium gradient during hydrogenotrophic methanogenesis whereas the latter i.e. methanogens with cytochromes have a dedicated electron transport chain as described in the text. Since the focus of the manuscript is on cytochromes, we would request the reviewer that we keep the classification scheme as is.

We acknowledge the reviewer's suggestion re: Figure S2 and have provided more details re: Mtr and Mcr in the revised figure S2.

Lines 173-178. The argument for the physiological role of MmcA in iron reduction is not necessary and detracts from the overall discussion. First, the obvious interpretation of Fig. 4 is that MmcA plays a minor or no role in iron anaerobic respiration is ignored. A 30% reduction in activity could easily have resulted from pleiotropic effects. Second, this experiment cannot rule out redundancy in the pathway, so the results of a single mutation are difficult to interpret. Third, reduction was measured and not anaerobic respiration, so it is also possible that the reduction was just a side reaction. Lastly, given the extensive washing that was performed on the cells, it would be nice to

compare the rates of iron reduction with those measured during growth with iron, which are available in the literature. This comparison would ensure that the major activity was being measured.

We have removed this paragraph per the reviewer's request. Please see lines 167-168 of the revised manuscript.

Lines 222-225. The claim that the manuscript demonstrates iron-dependent respiration is not supported by the evidence. Only iron reduction was measured. Evidence was not presented for generation of a proton motive force or ATP, which is required to claim respiration. The authors could delete this very speculative paragraph without detriment to the value of the paper.

We thank the reviewers for this comment. We have removed this sentence per the reviewer's request. Please see lines 210-211 of the revised manuscript.

Lines 241-250. This paragraph seems to be premature and could be deleted.

We have removed this paragraph per the reviewer's request. Please see lines 226-227 of the revised manuscript.

Line 203. A word seems to be missing in this sentence.

We have corrected this typo and thank the reviewer for bringing it to our attention. Please see line 191 of the revised manuscript.

Reviewer #5 (Remarks to the Author):

As an adjudicating reviewer, I have carefully considered how novel and rigorous this manuscript in the context of the three previous reviews. Overall, the manuscript represents a major advance as the first biochemical characterisation of a c-type cytochrome in methanogens, benefiting from strong activity studies and electrochemical analysis. I agree that, on the basis of the mild effects observed in the mutant data, the physiological role of this enzyme remains an open question. But I am personally satisfied that the manuscript makes a sufficient independent advance irrespectively and think the authors have responded appropriately to the reviewers.

We thank the reviewer for their insightful assessment and consideration.

REVIEWERS' COMMENTS

Reviewer #2 (Remarks to the Author):

The authors have addressed all comments I raised on the minor points, and have argued on the innovative and groundbreaking nature of their results, which I raised as a major concern. The minor concerns have been addressed appropriately.

For the major concerns, however, I am unable to change my opinion. It does not question the

validity of the results of the authors, but rather the impact, novelty and groundbreaking nature of their results. The authors argue that the biochemical function was previously not known, whereas in my opinion other experiments have already suggested this function (documented in numerous papers mentioned in my first review, e.g. by deletion mutant studies). I do not see the confirmation of previous experiments via a different experimental approach as groundbreaking, more as affirmatory - which is fine, but in my view does not warrant publication in Nature communications. They argue that this article is the first to investigate isolated c-type cytochromes in methanogenic archaea. Whereas this is undoubtedly true, I think the innovation doesn't come from doing something for the first time, but by the insights gained from this - and I couldn't find groundbreaking new insights resulting from the author's experiments. In this paper, results are either affirmatory of previous experiments or descriptive, which I think is not groundbreaking. The authors further argue that this opens the way to study ANME cytochromes. This may or may not be the case, so the innovation then would come from actually studying those cytochromes, not suggesting that it is possible to do so.

In essence, I'm curious how the editor will decide whether such results can be seen as highly innovative and groundbreaking, to be publishable in Nature Comm. In my opinion, the impact of the results of the authors are not warranting this.

Reviewer #3 (Remarks to the Author):

The authors have addressed my concerns.